# AN INFORMATION-THEORETIC APPROACH TO UNSUPERVISED KEYPOINT REPRESENTATION LEARNING

## ABSTRACT

Extracting informative representations from videos is fundamental for effective learning of various downstream tasks. Inspired by classical works on saliency, we present a novel information-theoretic approach to discover meaningful representations from videos in an unsupervised fashion. We argue that *local entropy* of pixel neighborhoods and its evolution in a video stream is a valuable intrinsic supervisory signal for learning to attend to salient features. We, thus, abstract visual features into a concise representation of keypoints that serve as *dynamic information transporters*. We discover in an unsupervised fashion spatio-temporally consistent keypoint representations, thanks to two original information-theoretic losses. First, a loss that maximizes the information covered by the keypoints in a frame. Second, a loss that optimizes transportation over time, imposing consistency of the information flow. We compare our keypoint-based representation to state-of-the-art baselines in different downstream tasks such as learning object dynamics. To evaluate the expressivity and consistency of the keypoints, we propose a new set of metrics. Our empirical results showcase the superior performance of our information-driven keypoints that resolve challenges like attendance to both static and dynamic objects, and to objects abruptly entering and leaving the scene.[1]

## 1 INTRODUCTION

Humans are remarkable for their ability to form representations of essential visual entities and store information to effectively learn downstream tasks from experience (Cooper, 1990; Radulescu et al., 2021). Research evidence shows that the human visual system processes visual information in two stages; first, it extracts sparse features of salient objects (Bruce & Tsotsos, 2005); second, it discovers the interrelations of local features for grouping them to find correspondences (Marr, 2010; Kadir & Brady, 2001). For videos with dynamic entities, humans not only focus on dynamic objects, but also on the structure of the background scene if this plays a key role in the information flow (Riche et al., 2012; Borji et al., 2012). Ideally, we want a learning algorithm to extract similar abstractions that can be useful to various downstream tasks. Notable research works in Computer Vision (CV) and Machine Learning (ML) have proposed different feature representations from pixels for challenging downstream tasks (Szeliski, 2010; Harris et al., 1988; Lowe, 2004; Rublee et al., 2011; Rosten & Drummond, 2006; Mur-Artal et al., 2015). Recent efforts focus on deep learning representations of Points of Interest (PoI) for tasks like localization and pose estimation (DeTone et al., 2018; Florence et al., 2018; Sarlin et al., 2020; Ono et al., 2018; Sarlin et al., 2019; Dusmanu et al., 2019).

Keypoints stand out as PoI with semantic interpretation (Jiang et al., 2009; Alexe et al., 2010), e.g., representing objects (Xiongwei et al., 2020) or the joints of a human pose (Kreiss et al., 2019) and can represent structure useful for learning control (Xiong et al., 2021). Many keypoint detection methods are trained in a supervised fashion, relying on annotations (Cao et al., 2017). Unsupervised and self-supervised learning methods can compensate for the need for expensive human annotations (Wang et al., 2019; 2020; Minaee et al., 2021; Kim et al., 2019; Yang et al., 2020; Gopalakrishnan et al., 2020; Chen et al., 2019). Current state-of-the-art methods for unsupervised keypoint discovery mainly focus on dynamic entities in a video (Kulkarni et al., 2019; Minderer et al., 2019), not effectively representing the scene's static and dynamic entities. Namely, these methods are trained to

---

[1]Project website: `https://sites.google.com/view/mint-iclr`

reconstruct differences between frames and cannot easily disambiguate occlusions or consistently represent randomly appearing-then-disappearing objects in a video stream.

This work introduces Maximum Information keypoiNTs (MINT), an information-theoretic treatment of keypoint-based representation learning by considering keypoints as the "transporters" of prominent information in a frame and subsequently through a video stream. Our proposed method relies on *local entropy* computed in neighborhoods (patches) around candidate keypoints. We argue that image entropy, and its changes over time, provide a strong *inductive bias* for training keypoints to represent salient objects, as early works in saliency detection pointed out (Kadir & Brady, 2001; Bruce & Tsotsos, 2005). To compute the entropy, we introduce a novel, efficient *entropy layer* that operates locally on image patches. MINT maximizes both the image entropy coverage by the keypoints and the conditional entropy coverage across frames. To do so, MINT relies on an original formulation of unsupervised keypoint representation learning with loss functions to *maximize the represented image entropy* and the *information transportation across frames* by the keypoints, imposing spatio-temporal consistency of the represented entities.

We provide qualitative and quantitative empirical results on four different video datasets that allow us to unveil the representation power of MINT against strong baselines of unsupervised keypoint discovery. Unsupervised keypoint representation learning is challenging to benchmark due to the absence of designated metrics and datasets. We, therefore, provide a new set of metrics with a downstream task in the domain of multiple object detection and tracking, based on CLEVRER (Yi et al., 2019). Moreover, we provide results on two challenging datasets (Sharma et al., 2018; Memmesheimer et al., 2019) that contain interesting dynamic scenes of various difficulties (close-up frames with dynamic interactions vs. high-res wide frames with clutter). We show that MINT economizes the use of keypoints, deactivating excessive ones when the information is well contained, and dynamically activates them to represent new entities entering the scene temporarily. Finally, to demonstrate the suitability of MINT as a representation for control, we devise an imitation learning downstream task based on the toy environments of the MAGICAL benchmark (Toyer et al., 2020).

To summarize, our contributions are: *(1)* an original information-theoretic approach to unsupervised keypoint representation learning that uses local image entropy as a training inductive bias, seeking to maximize the represented information in videos; *(2)* an entropy layer to compute local image entropy in patches; *(3)* an unsupervised way for learning to represent a variable number of entities in video streams by activating/deactivating keypoints to cover the necessary information; *(4)* a new set of evaluation metrics in a simple and intuitive downstream task for benchmarking the performance of unsupervised keypoint discovery methods.

## 2  MAXIMUM INFORMATION KEYPOINTS

We propose an unsupervised method for keypoint discovery in videos based on information-theoretic principles. Keypoints should adequately represent the scene and dynamic changes in it. Starting from the original assumption that a keypoint represents a patch of information on the image, we leverage local image entropy to measure the representation power of keypoints in terms of their transmitted amount of information. Consequently, we argue that keypoints should cover areas in the image that are rich in information, while the number of keypoints should dynamically adapt to represent new information. Finally, keypoints should consistently represent the same information pattern spatiotemporally in a video. With this motivation, we propose maximizing the information covered by the keypoint representation in a video through the introduction of two novel losses based on information-theoretic measures. *(1)* An information maximization loss encourages the keypoints to cover areas with high entropy in a single frame. *(2)* An information transportation loss enables the keypoints to represent the same entity over subsequent frames. In the following, we present these losses and theoretical analyses supporting their design.

### 2.1  PIXEL INFORMATION & LOCAL IMAGE ENTROPY

Our information-theoretic approach for unsupervised keypoint discovery requires quantifying the amount of information each pixel location in a single frame carries. We measure the information of a pixel via Shannon's entropy definition (Shannon, 2001), based on the probability of each pixel. Images can be considered lattices, with pixels being the random variables (Li, 2009). We compute the

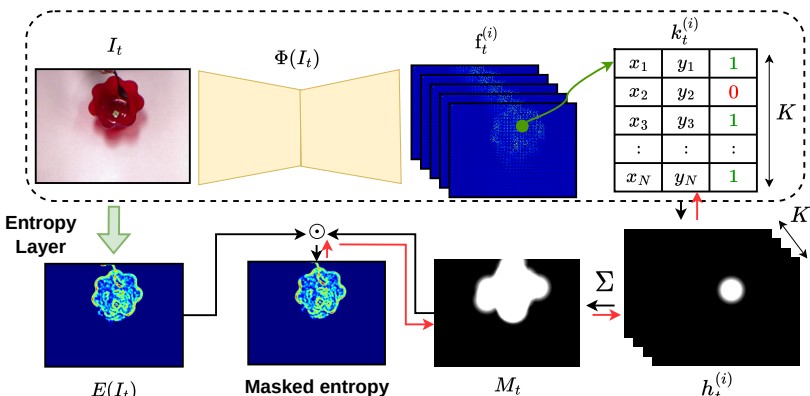

Figure 1: The architecture of our keypoint model $\Phi(I_t)$ (Sec. 2.2) and the masked entropy (Sec. 2.2.1). For an input image $I_t$ our model $\Phi(I_t)$ outputs $K$ feature maps $f_t^{(i)}$ for each keypoint $k_t^{(i)}$, $i \in \{1, \ldots, K\}$. A heatmap $h_t^{(i)}$ is generated for each keypoint, while the active keypoints are aggregated to form the mask $M_t$. The entropy layer computes the entropy of the image $E(I_t)$. Our masked entropy (ME) loss maximizes the percentage of the entropy in the masked entropy image. Red arrows show the backward gradient flow. Only the part encircled by the dashed line is used during inference.

discrete probability density of a pixel using the statistics of the color intensities in its neighborhood, represented by a normalized histogram of the neighboring pixel values (Sabuncu, 2006). In order to compute these histograms efficiently and to derive the final entropy image, we developed a computationally optimized entropy layer as detailed in Appendix B.

Our entropy layer estimates the pixel-wise entropy image $E(I)$ for an RGB input image $I \in \mathbb{R}^{H \times W \times 3}$, with $H$ being the height and $W$ the width of an image frame with 3 color channels. $E(I)$ consists of the local entropies $E(I(x, y))$ computed at each pixel location $(x, y)$ by estimating the entropy of the neighborhood region $R(x, y)$ centered at $(x, y)$, using a normalized histogram-based discrete probability function $p(b, R(x, y))$ for each color value $b$ in the region $R(x, y)$ summed and normalized over the color channels (details in Appendix B). The final per-pixel local entropy is

$$E(I(x, y)) = - \sum_{b \in [0, 255]} p(b, R(x, y)) \log(p(b, R(x, y))) . \tag{1}$$

Computing the entropy for all pixels results in the pixel-wise entropy image $E(I)$.

## 2.2 INFORMATION-THEORETIC KEYPOINT DISCOVERY

We consider the keypoints as a compact representation of images, which attend to salient entities in a scene (Szeliski, 2010). In our method, keypoints represent distinctive information patterns overlaid on a set of neighboring pixels in an image frame. We explicitly treat the keypoint as a transporter of information in its surrounding patch, and we develop an end-to-end approach for unsupervised keypoint discovery in videos, based on information-theoretic principles.

We define a keypoint discovery model $\Phi(I_t)$ (cf. Fig. 1), which is a deep neural network that discovers $K$ keypoints $k_t^{(i)}$, $i \in \{1, ..., K\}$, in an input color image $I$ at time $t$. It outputs $K$ feature maps $f_t^{(i)}$, each corresponding to one keypoint. The coordinates $(x_i, y_i)_t$ of the respective keypoint $k_t^{(i)}$ are obtained with a spatial soft-argmax (Levine et al., 2016). Besides predicting the coordinates, the model also assigns an activation status $s_t^{(i)} = \{0, 1\}$ per keypoint. The activation status determines whether a keypoint is active ($s_t^{(i)} = 1$) or not ($s_t^{(i)} = 0$) in a specific frame, allowing the network to learn to decide on the ideal number of active keypoints. Overall, a keypoint is defined by its coordinates and the activation score $k_t^{(i)} = (x_i, y_i, s^{(i)})_t$. Additionally, we define a differentiable heatmap $h_t^{(i)} \in \mathbb{R}^{H \times W}$ for each $i^{\text{th}}$ keypoint by thresholding a distance-based Gaussian $G_t^{(i)}$ centered at the coordinates of the keypoint (details in Appendix A.3). The heatmaps localize the keypoint information coverage. We want to maximize the coverage of information by the keypoints spatio-temporally in the video stream. For that, we need to ensure that both the per frame and intra-frame information is sufficiently transmitted by the keypoints. Inspired by information theory, we derive novel losses that allow us to learn information-driven keypoint representations, while providing error bounds that theoretically justify the design of those losses (Sabuncu, 2006; Yu et al., 2021).

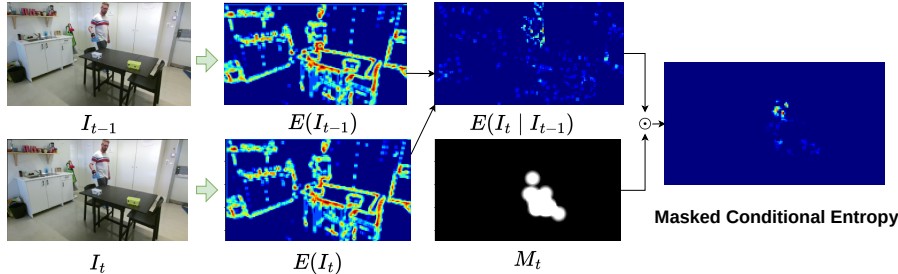

Figure 2: Masked conditional entropy (MCE) computation. Given two consecutive images $I_{t-1}$ and $I_t$, we extract their entropy images $E(I_{t-1})$ and $E(I_t)$. The conditional entropy $E(I_t|I_{t-1})$ depends on the entropy image of both images. Multiplying the conditional entropy image by the aggregated mask $M_t$ gives the MCE image. The masked conditional entropy (MCE) loss maximizes the percentage of the masked conditional entropy.

### 2.2.1 MAXIMIZATION OF KEYPOINT INFORMATION

With information maximization, we encourage keypoints to represent image regions rich in information. We leverage local image information entropy (Sec. 2.1) to define losses for keypoint discovery. We want to enforce maximum collective information coverage by the keypoints in a frame for representing all entities in a scene. For that, we introduce two losses; the masked entropy (ME) loss and the masked conditional entropy (MCE) loss.

The **ME loss** encourages the information coverage by the keypoints in a single frame. We use the heatmap $h_t^{(i)}$ of each keypoint $k_t^{(i)}$ to retrieve the local image information at time $t$, We filter out inactive keypoints by multiplying the heatmap with the activation status $s_t^{(i)}$. Aggregating all heatmaps gives the aggregated mask $M_t = \min(\sum_i^K h_t^{(i)} \odot s_i^{(t)}, 1)$.

Considering keypoints as channels of information, we arrive at the following theorem that bounds the information loss by the keypoints' masking of the original image. The bound follows Fano's inequality (Sabuncu, 2006; Scarlett & Cevher, 2019), and proves that maximizing the keypoints' masked entropy lowers the probability of error of the information loss by this keypoint representation.

**Theorem 1.** *Let $I_t^M$ be the masked image at time $t$, obtained by the operation $I_t^M = I_t \odot M_t$, where $\odot$ denotes the Hadamard (i.e., element-wise) product. Let $\mathcal{B}$ be the "vocabulary" of pixel intensities, and we assume that every pixel in location $(x, y)$ is uniform on $\mathcal{B}$. The average error probability $\bar{P}_\varepsilon$ over all pixels $N = H \times W$ of the information approximated by $I_t^M$ w.r.t. to the original image $I_t$ can be lower bounded by*

$$\bar{P}_\varepsilon \geq 1 - \frac{\sum_{x,y}(E(I_t^M(x,y)))}{N \log |\mathcal{B}|} - \frac{\log 2}{\log |\mathcal{B}|} . \tag{2}$$

Proof in Appendix C. We can assume that the upper bound for the error probability remains 1, because of the activation $s$ of the keypoints, there is a probability that the masked image is "empty", i.e., all keypoints inactive. From Eq. (2) we can see that the ME maximization lowers the probability of error. This motivates the practical implementation of the ME loss $\mathcal{L}_{ME}(I_t)$ that optimizes the percentage of the masked entropy over the all pixel locations $(x, y)$, $\sum_{x,y} E(I_t) \odot M_t$ w.r.t. the total image entropy $\sum_{x,y} E(I_t)$

$$\mathcal{L}_{ME}(I_t) = 1 - \frac{\sum_{x,y} E(I_t) \odot M_t}{\sum_{x,y} E(I_t)} = 1 - \frac{\sum_{x,y} E(I_t) \odot \min(\sum_{i=1}^K h_t^{(i)} \odot s_t^{(i)}, 1)}{\sum_{x,y} E(I_t)} . \tag{3}$$

The **MCE loss** encourages the keypoints to pay special attention to dynamic entities when the available number of keypoints are insufficient for covering the information in a sequence of frames. The conditional entropy of an image $I_t$ at time $t$ given a reference image $I_{t-1}$ at time $t - 1$ measures the information change of pixels, indicating moved objects. Optimizing the conditional entropy $E(I_t|I_{t-1})$ in a sequence of images encourages the keypoint detector to attend to moving objects (cf. Fig. 2). The pixel-wise conditional entropy of an image can be computed by the joint entropy of two images subtracting the reference image entropy $E(I_t|I_{t-1}) = E(I_t, I_{t-1}) - E(I_{t-1})$, where $E(I_t, I_{t-1}) \approx \max(E(I_t), E(I_{t-1}))$ following Lemma 1.

**Lemma 1.** *The joint entropy of two images $I_1$ and $I_2$ can be approximated by $E(I_1, I_2) \approx \max(E(I_1), E(I_2))$.*

Accordingly, we can bound the information loss by the keypoints in a sequence of frames.

**Corollary 1.** *Following Theorem 1, we can bound the average probability of error $\bar{P}_\varepsilon^{cond}$ of the conditioned masked images in time as*

$$\bar{P}_\varepsilon^{cond} \geq 1 - \frac{\sum_{x,y} E(I_t^M(x,y)|I_{t-1}^M(x,y))}{N \log |\mathcal{B}|} - \frac{\log 2}{\log |\mathcal{B}|} \ . \tag{4}$$

Following Eq. (4), we observe that the MCE maximization lowers the probability of error of the conditional information loss between frames, leading to the practical implementation of the MCE loss, similarly to the ME loss, $\mathcal{L}_{MCE}(I_t, I_{t-1})$ that maximizes the percentage of total masked conditional entropy $\sum_{x,y} E(I_t|I_{t-1}) \odot M_t$ to the total conditional entropy $\sum_{x,y} E(I_t|I_{t-1})$

$$\mathcal{L}_{MCE}(I_t, I_{t-1}) = 1 - \frac{\sum_{x,y} E(I_t|I_{t-1}) \odot M_t}{\sum_{x,y} E(I_t|I_{t-1})} \ . \tag{5}$$

### 2.2.2 MAXIMIZATION OF KEYPOINT INFORMATION TRANSPORTATION

Keypoints should transmit information about the same entity over time. Temporal consistency means aligning each keypoint to the same information pattern across its occurrences. Parting from this idea and inspired by feature transportation (Kulkarni et al., 2019), we propose information transportation (**IT**), relying on image entropy and waiving the need for image reconstruction.

In a temporal sequence of frames, we can perform keypoint information transportation by *reconstructing the image information entropy* of the current frame $E(I_t)$ using information from previous frame $E(I_{t-1})$ (cf. Fig. 3). The conditional image entropy of the two frames $E(I_t|I_{t-1})$, represents the amount of pixel-wise information needed to describe the information of $E(I_t)$ given $E(I_{t-1})$. Let's consider the $i^{\text{th}}$ keypoint at time step $t$ (coordinates are omitted for avoiding verbosity). Its associated heatmap $h_t^{(i)}$ represents the spatial information covered by the $i^{th}$ keypoint. We can construct a *source information image* $E(S_t^{(i)})$ by subtracting the information carried by the keypoint in frames $t-1$ and $t$ from the information image $E(I_{t-1})$, i.e., $E(S_t^{(i)}) = E(I_{t-1}) \odot (1-h_{t-1}^{(i)}) \odot (1-h_t^{(i)})$. Implanting the keypoint information represented by $h_t^{(i)}$ onto the conditional image entropy $E(I_t|I_{t-1})$ that contains all conditional information except for the one of the $i^{\text{th}}$ keypoint forms the target pixel-wise information $E(T_t^{(i)}) = E(I_t) \odot (h_t^{(i)}) + \kappa E(I_t|I_{t-1}) \odot (1 - h_t^{(i)})$.[2] The reconstruction of the image information $E(I_t)$ results from the sum of the source and target image information $E(R_t^{(i)}) = E(S_t^{(i)}) + E(T_t^{(i)})$. This reconstruction pro-

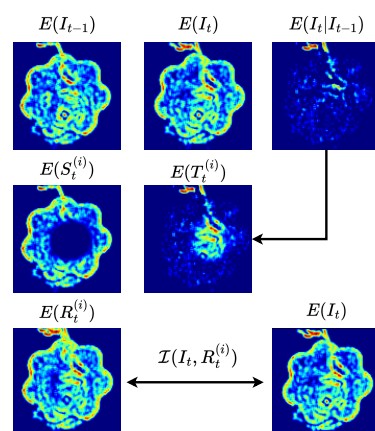

$E(I_{t-1})$  $E(I_t)$  $E(I_t|I_{t-1})$

$E(S_t^{(i)})$  $E(T_t^{(i)})$

$E(R_t^{(i)})$  $E(I_t)$

$\mathcal{I}(I_t, R_t^{(i)})$

Figure 3: Information transportation for a keypoint $k^{(i)}$. The source is the entropy of the previous image $E(I_{t-1})$ and the target is the entropy of the current frame $E(I_t)$. Removing the patch around the keypoint position at time $t-1$ and $t$ gives the source information $E(S_t^{(i)})$. The sum of the conditional entropy $E(I_t|I_{t-1})$ and the patch around the keypoint at time $t-1$ gives the target information $E(T_t^{(i)})$. The reconstructed information after transportation is the sum of target and source information $E(R_t^{(i)})$. The objective is to maximize the mutual information between the reconstructed entropy and the entropy of the current frame $\mathcal{I}(I_t, R_t^{(i)})$.

cess leads us to the following theorem, that shows that maximizing the mutual information (MI) between the per keypoint reconstructed information and the original image entropy, lowers the probability of error due to information loss.

**Theorem 2.** *Following Fano's inequality (Sabuncu, 2006; Scarlett & Cevher, 2019), we prove that the average error probability of the transportation for the $i^{th}$ keypoint $P_\varepsilon^{IT(i)}$ is lower bounded by*

$$\bar{P}_\varepsilon^{IT(i)} \geq 1 - \frac{\sum_{x,y} \mathcal{I}(I_t(x,y), R_t^{(i)}(x,y))}{N \log |\mathcal{B}|} - \frac{\log 2}{\log \mathcal{B}} \ . \tag{6}$$

---

[2] The factor $\kappa \leq 1$ encourages the network to concentrate on transportation more than on reconstruction.

*Considering all transportations independent, we can bound the joint average error probability, by summing the per keypoint error probability*

$$\bar{P}_\varepsilon^{IT\,(joint)} \geq K - \frac{\sum_{i=1}^{K} \sum_{x,y} \mathcal{I}(I_t(x,y), R_t^{(i)}(x,y))}{N \log |\mathcal{B}|} - \frac{K \log 2}{\log \mathcal{B}}. \tag{7}$$

Proof in Appendix C. From Eq. (6), we deduce that for optimizing the $i^{\text{th}}$ keypoint's information transportation, we should maximize the mutual information (MI) $\mathcal{I}(I_t, R_t^{(i)})$. This motivates our practical implementation of the information transportation (IT) loss for all keypoints. We construct the IT loss through the difference $E(I_t) - \mathcal{I}(I_t, R_t^{(i)})$ normalized by the area of the heatmap $A_h$ (equal for all keypoints). Minimizing $E(I_t) - \mathcal{I}(I_t, R_t^{(i)})$ maximizes MI, as dictated by Theorem 2. We found that normalizing with $A_h$ helps in practice to have a better scale. We, also, regularize the excessive keypoints' movement by minimizing the norm of the distance traveled by each keypoint $d_t^{(i)} = ||(x_i, y_i)_t - (x_i, y_i)_{t-1}||_2^2$ (scaled by a weight $m^d$). The final IT loss becomes

$$\mathcal{L}_{\text{IT}}(I_t, I_{t-1}) = \sum_{i=1}^{K} \frac{E(I_t) - \mathcal{I}(I_t, R_t^{(i)})}{A_h} + m_d \cdot d_t^{(i)}. \tag{8}$$

### 2.2.3 THE MINT LOSS & AUXILIARY LOSSES

The **overlapping loss** provides an auxiliary supervisory signal that spreads the keypoint over the image, encouraging them to cover distinctive regions. The sum of the Gaussians $G_t^{(i)}$ (cf. Appendix A.3) around the keypoints $k_t^{(i)}$ help estimate their overlap, $\mathcal{L}_o = min(max(\sum_i^K G_t^{(i)}) - \beta, 0)/K$. The overlapping loss minimizes the maximum of the aggregation of the Gaussian normalized by the number of keypoints $K$ with a lower bound $\beta$ to allow occlusion and avoid over-penalization.
The **active status loss** encourages the model to deactivate unnecessary keypoints, by minimizing the normalized sum of active keypoints while maximizing the ME. The interplay of the losses allows the method to eventually reach a trade-off between the number of active keypoints and the covered entropy. The active status loss optimizes $\mathcal{L}_s = (\sum_i^K s_t^{(i)})/K$.
The overall **MINT loss** $\mathcal{L}_{MINT}$ is a weighted combination of all losses (with dedicated weight $\lambda$ per loss), with the weight of the status loss reversed to schedule it according to the percentage of ME

$$\mathcal{L}_{MINT} = \lambda_{ME}\mathcal{L}_{ME} + \lambda_{MCE}\mathcal{L}_{MCE} + \lambda_{IT}\mathcal{L}_{IT} + \lambda_o\mathcal{L}_o + (1 - \mathcal{L}_{ME})\lambda_s\mathcal{L}_s. \tag{9}$$

Further information about the hyperparameters are available in Appendix E.2.

## 3 EXPERIMENTS

We evaluate MINT on four datasets ranging from videos of synthetic objects – CLEVRER (Yi et al., 2019) and MAGICAL (Toyer et al., 2020) – to realistic human video demonstrations – MIME (Sharma et al., 2018) and SIMITATE (Memmesheimer et al., 2019). Our experiments show the efficacy of our method as a representation for different tasks, and we provide quantitative results w.r.t. evaluation metrics (for object detection and tracking on CLEVRER) for several downstream tasks (learning dynamics on CLEVRER, imitation learning on MAGICAL), and qualitative results on the challenging datasets of MIME and SIMITATE. We compare against baselines for unsupervised end-to-end keypoint detection **Transporter** (Kulkarni et al., 2019), **Transporter-modified**, i.e., a modified version with a smaller receptive field, **Video Structure** (Minderer et al., 2019), an ablations of MINT; **MINT w/o Reg.**, i.e., MINT without the regularization terms, and an end-to-end CNN-based feature extraction. We report statistics for all quantitative results over 5 seeds. An extensive ablation study of MINT is provided in Appendix E.1 and baselines are discussed in Appendix E.3.

**Downstream task I: *Object detection and tracking*.** Capturing scene structure requires detecting all objects in an image, while object tracking is essential for representing the scene's dynamics. MINT can successfully train a spatio-temporally consistent keypoint representation on videos, leading to its natural application for object (static/dynamic, appearing/disappearing) detection and tracking.
We use CLEVRER (Yi et al., 2019), a dataset for visual reasoning with complete object annotations, containing videos with static and dynamic objects, with good variability in the scenes, as a testbed. To

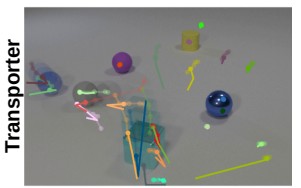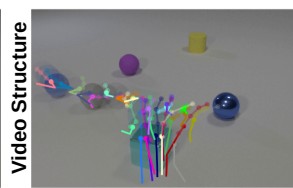

Figure 4: Qualitative results on CLEVRER dataset for Task I (object detection and tracking) and Task II (learning dynamics). Our method is able to assign keypoints to all objects, independent of whether they move or not, and follows their trajectory. The number of keypoints is dynamically adjusted to the number of objects. Future states are transparent, as well as the predicted keypoint and trajectories.

Table 1: Quantitative evaluation of keypoint detection and tracking on CLEVRER (Yi et al., 2019).

| Method | DOP ⇑ | TOP ⇑ | UAK ⇓ | RAK ⇓ |
|---|---|---|---|---|
| MINT w/o Reg. (ours) | **0.918 ± 0.073** | **0.897 ± 0.078** | 6.793 ± 1.956 | 2.478 ± 0.865 |
| MINT (ours) | *0.855 ± 0.118* | *0.838 ± 0.121* | **0.889 ± 0.639** | **1.123 ± 0.448** |
| Transporter | 0.787 ± 0.113 | 0.745 ± 0.119 | 18.417 ± 1.639 | 1.157 ± 0.323 |
| Transporter-modified | 0.832 ± 0.107 | 0.794 ± 0.114 | 16.267 ± 2.349 | 1.764 ± 0.671 |
| Video Structure | 0.567 ± 0.256 | 0.543 ± 0.253 | 18.104 ± 3.538 | 1.922 ± 0.652 |

Table 2: Prediction success rate on CLEVRER (Yi et al., 2019).

| Method | 1-step prediction | 2-steps prediction | 3-steps prediction |
|---|---|---|---|
| MINT (ours) | **0.844 ± 0.116** | **0.827 ± 0.126** | **0.811 ± 0.132** |
| Transporter | 0.746 ± 0.116 | 0.716 ± 0.120 | 0.692 ± 0.122 |
| Transporter-modified | 0.814 ± 0.099 | 0.791 ± 0.106 | 0.769 ± 0.110 |
| Video Structure | 0.734 ± 0.124 | 0.719 ± 0.125 | 0.699 ± 0.127 |

quantitatively assess the performances of MINT, we developed evaluation metrics for CLEVRER. We propose the **percentage of the detected object (DOP)** and the **percentage of tracked objects (TOP)** as two metrics, with higher values corresponding to better keypoint detection and tracking. A keypoint detects an object if it lies on its mask, and tracks it, if it detects the same object in two consecutive frame. Assigning keypoints to areas already represented by other keypoints or empty spaces signals bad keypoint detection. To evaluate these cases, we define two additional metrics for the **redundant keypoint assignment (RAK)** and **unsuccessful keypoint assignment (UAK)**, with lower values corresponding to better detection. The metrics are described in detail in Appendix D. We train all keypoint detectors on a subset of 20 videos from the CLEVRER dataset and test them on 100. The train-test split emulates a low-data regime and tests the methods' generalization abilities. As seen in Table 1, MINT w/o Reg. detects more objects (DOP) and tracks them better (TOP), showing the benefit of our information-theoretic losses. The proposed MINT model exhibits the best trade-off between superior performance against the baselines on all metrics, and better handling of keypoint assignment (UAK and RAK) than MINT w/o Reg. See more discussion about the ablations in Appendix E.1. These results are supported by the visual comparisons in Fig. 4, and videos on the project website[3].

**Downstream task II: *Learning dynamics*.** Proper object detection allows us to learn the underlying dynamics that evolve a scene. We test the representation power of the discovered keypoints by training a prediction model using the pre-trained keypoint detectors from Task I (using the best seed for each method). The prediction model treats the keypoints as graph nodes in an Interaction Network (IN) (Battaglia et al., 2016) to model the relational dynamics (cf. Appendix E.4). We train the prediction model to forecast the future positions of the keypoint given a history window of four-time steps. We compare the prediction against the ground truth position of the object in the predicted frame using CLEVRER (Yi et al., 2019). We report in Table 2 the ratio of successfully predicted objects (i.e., a predicted keypoint lying on the same object in the next frame) to the ground truth number of objects in the next time step. The comparison demonstrates that keypoints detected by our method represent the scene better than the baselines and help to predict the next state. Fig. 4 shows the prediction performance using different keypoint detectors.

**Downstream task III: *Object discovery in realistic scenes*.** Our method addresses challenging aspects beyond synthetic datasets. We evaluate the keypoint detectors on two additional datasets; (1) MIME (Sharma et al., 2018): a collection of close-up videos of human hands manipulating objects, and (2) SIMITATE (Memmesheimer et al., 2019): a video dataset of humans performing manipulation

---

[3]Videos at `https://sites.google.com/view/mint-iclr` or in the supplemental material.

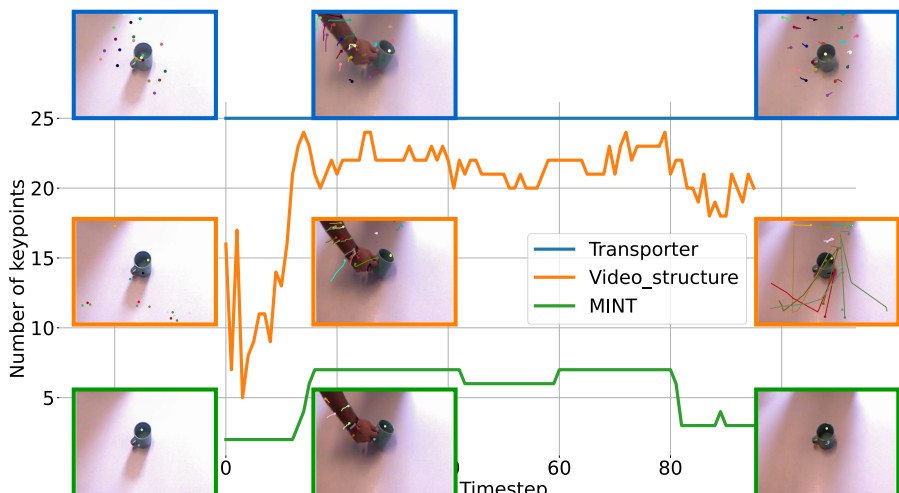

Figure 5: Come-and-go scenario. In a manipulation video from MIME, the hand enters after the start of the video and departs before the end. We plot the number of active keypoints w.r.t. timesteps. Transporter has a fixed number of keypoints. Video structure increases the number of active keypoints when the hand appears, but struggles when it disappears. MINT activates and deactivates the suitable number of keypoints.

tasks in wide-view cluttered scenes. For training, we use a subset of 80 videos from various tasks, and for testing a 100, and since no annotations are provided, we rely on qualitative analysis.

In MIME, the human hand enters and leaves the scene abruptly, allowing the evaluation of MINT in come-and-go scenarios shown in Fig. 5. MINT activates only the required number of keypoints, while Transporter (Kulkarni et al., 2019) uses a static number of keypoints, and Video structure (Minderer et al., 2019) fails to deactivate the excessive keypoints when the hand disappears. Fig. 5 shows the number of active keypoints over time, revealing our method's superior performance, both in keypoints number economy and in the qualitative representation of the objects in the scene.

The qualitative results for SIMITATE, on the other hand, in Fig. 6 show that only MINT can disambiguate between static and dynamic objects, tracking the human movement, while maintaing the structure of the keypoints relatively constant over the static objects. The baselines rely on reconstructing the movement, failing to represent the scene's structure. The qualitative results in SIMITATE reveal the need for the conditional entropy loss (forcing attention on moving objects when number of available keypoints is restricted) and the information transportation loss (ensuring the spatio-temporal consistency) – see also Fig. 10 in Appendix.

**Downstream task IV: *Imitation learning*.** Imitation learning from video frames is a long-standing challenge for control. Keypoints can define a low-dimensional representation, that could reduce the computational burden considerably. In this experiment, we investigate the suitability of our keypoint representation for control tasks, like imitation learning in MAGICAL (Toyer et al., 2020). We first pretrain MINT on 24 demonstration videos from different tasks. Then, we fix the keypoint detector and train an agent to mimic the demonstrated actions, using an IN (Battaglia et al., 2016), followed by a fully-connected layer that decodes the actions (cf. Appendix E.4). The agent uses as input the observed keypoints from four frames. We also found it useful to predict the next state as an auxiliary task. We compare the MINT-based agent against an agent that uses a CNN to extract features directly from pixels. The CNN agent is trained from scratch for each environment (cf. Appendix E.5). We consider three environments with different levels of difficulty; **MoveToRegion**: move an agent to a specific region, only the agent is involved (easy). **MoveToCorner**: move an object to the top-left corner, one object and the agent are involved (medium). **MakeLine**: place multiple objects in a line, four objects and the agent are involved (hardest). We evaluate the learned policy on environment instances from demonstrations (Demo) and randomly initialized (TestJitter). The results in Table 3 reveal that a pretrained keypoint model with MINT is suitable for control, achieving comparable or even superior performance to a task-specific CNN-based agent (cf. Appendix E.5 for more details).

**Limitations.** Our method relies on local image entropy after filtering high-frequency color changes. As a result, the method has difficulties in recognizing transparent objects and objects with the same color as the background. We plan to investigate the integration of implicit representation learning to counterpart this issue. Another limitation is the interpretation of the keypoints in the three-dimensional

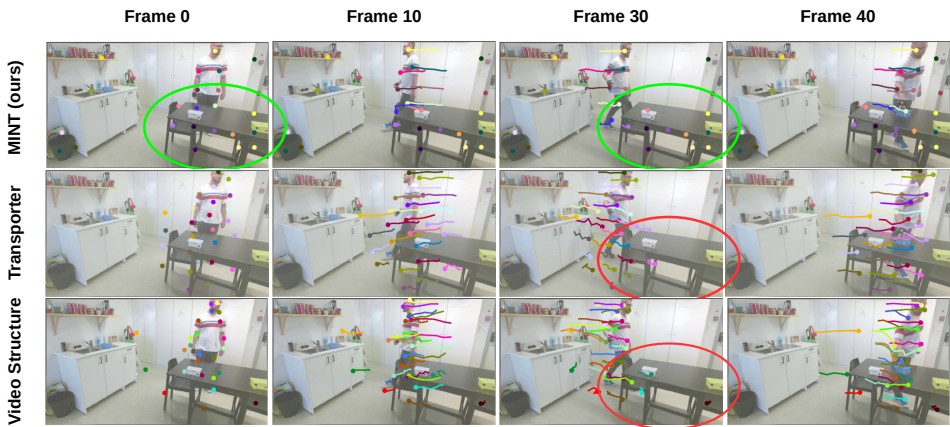

Figure 6: Crowded scenes. A video from SIMITATE dataset with a human moving in a room. All the methods can track the human successfully, but only MINT (ours) can keep the keypoints on the static objects consistently (green ellipses), while the baselines lose track of them (red ellipses).

Table 3: Average score for imitation learning on MAGICAL (Toyer et al., 2020). Higher values are better.

| Method | MoveToRegion | | MoveToCorner | | MakeLine | |
| --- | --- | --- | --- | --- | --- | --- |
| | Demo | TestJitter | Demo | TestJitter | Demo | TestJitter |
| MINT (ours) | **1.00 ± 0.00** | **0.86 ± 0.31** | **1.00 ± 0.00** | **0.80 ± 0.34** | **0.2 ± 0.22** | 0.06 ± 0.14 |
| CNN | **1.00 ± 0.00** | 0.84 ± 0.32 | 0.74 ± 0.35 | 0.30 ± 0.38 | 0.00 ± 0.00 | 0.01 ± 0.06 |

space. The current method operates on images and does not provide 3D information. Adding depth information or extending to a multi-view setting are options for future improvements.

## 4 RELATED WORK

Our work draws inspiration from classical approaches for saliency detection in images, using local information to detect salient objects (Kadir & Brady, 2001; Bruce & Tsotsos, 2005; Fritz et al., 2004; Renninger et al., 2004; Borji & Itti, 2012). The idea of extracting sparse feature representations of high-dimensional visual data is dominant in computer vision (Harris et al., 1988; Lowe, 2004; DeTone et al., 2018). Traditional geometric computer vision methods relied on the extraction of hand-crafted feature descriptors (Lowe, 2004; Rublee et al., 2011; Schmid et al., 2000; Mur-Artal et al., 2015). Recently, CNN architectures have proven superior (Yi et al., 2016; DeTone et al., 2018; Song et al., 2020; Zheng et al., 2017) performance. Keypoints represent a class of PoI that have a semantic entity (Duan et al., 2019; Cao et al., 2017; McNally et al., 2022), but most methods rely on explicit annotations of keypoint locations for training. Related to our work are methods that rely on image reconstruction for detecting keypoints in video streams in an unsupervised fashion (Jakab et al., 2018; Zhang et al., 2018b; Minderer et al., 2019; Kulkarni et al., 2019; Gopalakrishnan et al., 2020). An extended related work discussion is provided in Appendix G.

## 5 CONCLUSION

We presented MINT, an information-theoretic approach for unsupervised keypoint discovery from videos. Treating keypoints as transporters of information, we defined two losses inspired by information theory; an information maximization loss and an information transportation loss. The losses promote covering areas with high information while ensuring spatio-temporal consistency. Additionally, the method learns to activate keypoints upon necessity and reduces redundant keypoints thanks to auxiliary losses. The experimental evaluation demonstrated the superior performance of our method for various downstream tasks, ranging from object detection to dynamics prediction and imitation learning. Moreover, we showed qualitatively that our method can tackle many challenges for keypoint detection in realistic scenarios, such as attending to static and dynamic objects and handling come-and-go situations in realistic videos. Overall, we demonstrated how to learn reasonable keypoint representations purely unsupervised from videos, with promising results for future applications.

## REPRODUCIBILITY STATEMENT

We aim to guarantee the reproducibility of our work by providing additional details of the method and the conducted experiments in the appendix. Moreover, we provide the code in the supplementary material, and we will make it public upon acceptance.

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

## APPENDIX

The appendix provides additional details on the architecture, the entropy layer, proofs, the description of evaluation metrics, experimental details with an ablation study, additional results and discussions, as well as comments on the used code. Moreover, we provide video results for better visualization and the code in the supplemental material.

# A  ARCHITECTURE

This section contains additional information about the model architecture in Sec. 2.2, implementation details on how to get the keypoint coordinates from the feature map, and the heatmaps for keypoints to ensure reproducibility.

## A.1  KEYPOINT MODEL

Keypoints provide a low-dimensional representation of high-dimensional RGB images. Therefore, many keypoint detection techniques are inspired by the autoencoder architecture (Goodfellow et al., 2016) which uses the bottleneck to consolidate the information into a reduced dimensionality for keypoint extraction (Minderer et al., 2019; Kulkarni et al., 2019). Instead, we suggest taking an hourglass architecture (Newell et al., 2016) which upscales the compressed information again and outputs several feature maps with high activation in places with eminent information (Ewerton et al., 2021; Xu & Takano, 2021; Newell et al., 2016). This allows the network to predict the information at the original image size yielding finer resolution and correspondence to the original pixels.

Our keypoint detector consists of an hourglass convolutional neural network with three convolutional layers, with kernel sizes of $5, 3, 3$ and strides of $3, 2, 2$, respectively. The upsampling part of the model consists of three transposed convolutional layers, with kernel sizes of $3, 3, 3$ and strides of $1, 2, 2$. The number of input and output channels for each layer depends on the number of keypoints $K$ and the number of input image channels $C$, see Table 4. The result is passed through a softplus layer to ensure the positivity of the feature maps. Lastly, we append a spatial softargmax layer (see Appendix A.2) to get the coordinates of the keypoints from the feature maps $f_i$. We initialize all the convolutional layers with Xavier's normal initialization (Glorot & Bengio, 2010) and add a leaky ReLU activation and a batch normalization layer after each of them. The total number of the parameters is $58,725$ for the input of size $320 \times 420$. We normalize the input to the range $[-0.5, 0.5]$.

Table 4: Architecture details for an RGB-image of $320 \times 480$ and $K = 25$ keypoints, there is a leaky ReLU layer and a BatchNorm2d layer (50 parameters) after each convolutional layer.

| layer (type) | Input channels | Output channels | kernel size | stride | Output shape | num_params |
|---|---|---|---|---|---|---|
| Normalize-1 | C | C | - | - | [3, 320, 480] | 0 |
| Conv2d-1 | C | K | 5 | 3 | [25, 79, 106] | 1,900 |
| Conv2d-2 | K | K | 3 | 2 | [50, 26, 35] | 5,650 |
| Conv2d-3 | K | 2K | 3 | 2 | [50, 26, 35] | 11,300 |
| ConvTranspose2d-1 | 2K | 2K | 3 | 1 | [50, 53, 71] | 22,550 |
| ConvTranspose2d-2 | 2K | K | 3 | 2 | [25, 107, 143] | 11,275 |
| ConvTranspose2d-3 | K | K | 3 | 2 | [25, 107, 143] | 5,650 |
| Softplus-14 | K | K | - | - | = | 0 |
| SpatialSoftargmaxLayer-15 | K | K | - | - | [25, 2] | 0 |
| Final output | | | | | [25,3] | total = 58,725 |

## A.2  FEATURE MAP TO KEYPOINT

The argmax operator is not differentiable, so we opted to use a differentiable spatial soft argmax as an alternative to extract the keypoints coordinates from the feature maps. The spatial soft argmax (Levine et al., 2016) takes $K$ 2D feature maps $f^{(i)}$, flattens the feature maps and compute for each of them the weights $\omega_i$

$$w_i = \frac{e^{f^{(i)} - max(f^{(i)})}}{\sum e^{f^{(i)} - max(f^{(i)})}} \ . \tag{10}$$

Before applying the softmax, we subtract the maximum value from the input, which does not change the output of the softmax but helps for numerical stability. In order to map the weights to coordinates, we generate a mesh grid $(x_{grid}, y_{grid})$ of $x$ and $y$ coordinates, with the same size as the input image. We flatten the mesh grid and compute the expected coordinates $[\hat{x}_i, \hat{y}_i]$ as the weighted sum of the coordinate grid with $\omega_i$. This process is visualized in Fig. 7.

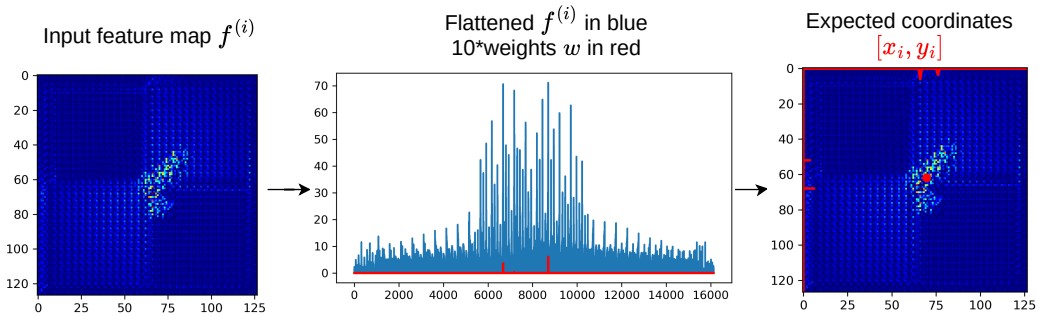

Figure 7: Feature map to keypoint (spatial-soft argmax)

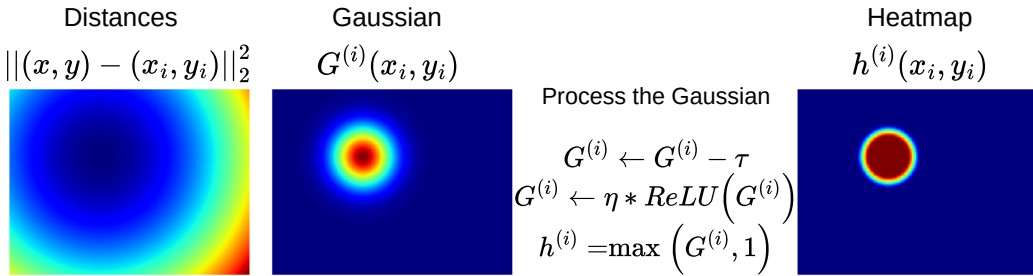

Figure 8: Keypoints to heatmap (Gaussian around keypoints)

### A.3 KEYPOINTS TO HEATMAPS

The heatmap $h^{(i)}$ generation for a keypoint takes coordinates as a pair of real numbers $(x_i, y_i) \in \mathbb{R}^2$. We start by generating a pixel-coordinates array with the same width and height as the original image $H \times W \times 2$, where 2 denotes the coordinates of each pixel $(x, y) \in \mathbb{N}^{H \times W \times 2}$. Then we compute the squared distance between the input and all the pixels $||(x, y) - (x_i, y_i)||_2^2$. We use the squared distance to generate a Gaussian distribution around the input coordinates $G^{(i)}(x_i, y_i)$ with a standard deviation $\sigma_{G^{(i)}}$.

The heatmap defines the area and weighting of information belonging to each keypoint. The heatmap should be 1 around the center of the keypoint, as the keypoint covers the information in this point completely, and descend gradually to 0 representing information out of reach of the keypoint. We achieve that by thresholding and clamping the Gaussian. We use a threshold $\tau$ and a scaling factor $\eta$ for the thresholded Gaussian to get the final heatmap $h_i$. Table 6 for more details on the scale of these hyperparameters. The process is visualized in Fig. 8.

## B ENTROPY LAYER

The entropy layer (Sec. 2.1) is the one of the main modules of our method. For an input image, the entropy layer outputs the entropy image. In this section, we provide additional implementation details. We split the explanation into two subsections: *(1)* the entropy module definition in PyTorch (Paszke et al., 2019), and, *(2)* the CUDA extension for the parallel execution.

### B.1 ENTROPY MODULE

The input of the entropy module is an RGB image $I \in \mathbb{R}^{H \times W \times 3}$. The input image is processed to remove high-frequency color changes. The processing is completely vectorized to allow efficient execution using PyTorch during training. The processing of the input image starts by blurring the image using an average blur layer, followed by sharpening the result, and finally dividing the sharp image by the smooth image. Fig. 9 reflects an example of the processing inside the entropy layer.

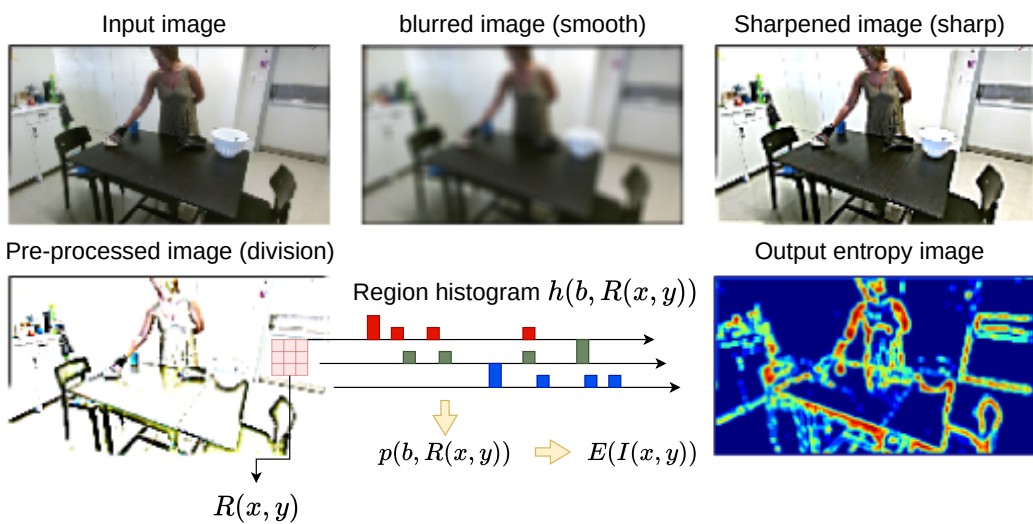

Figure 9: Entropy layer with the prepossessing

Before forwarding the processed image into the entropy function, we generate the neighborhood region $R(x, y)$ for each pixel location $(x, y)$. These regions have square shape with the corresponding pixel $(x, y)$ being the center. Instead of iterating over all the pixels with for loops, we use the stridden operation to factorize the extraction of the regions. Using strides to extract the regions aligns with how the image is stored in the memory and does not create overhead. These tricks are essential for the efficiency of our entropy layer.

### B.2 CUDA EXTENSION

We need the histogram of the color values to compute the entropy. However, the naive implementation via vectorizing the code requires computing a pairwise distance-matrix between each pixel with every histogram bin, which means multiplying the number of the possible regions by 256. This causes exploding GPU memory requirements (more than 50GB). Motivated by this observation, we present in this work an efficient entropy layer based on kernel density estimation.

To estimate the value for each histogram bin $b$ inside a region $R(x, y)$ centered on pixel at location $(x, y)$, we use the kernel density estimator

$$\hat{f}(b, x, y) = \sum_{(x_n, y_n) \in R(x,y)} \mathcal{K}(\frac{I(x_n, y_n) - b}{B}) \,, \tag{11}$$

where $I(x_n, y_n)$ is the pixel value at location $(x_n, y_n)$ in the region $R(x, y)$, and $B$ is the bandwidth, used as a smoothing parameter.
We follow (Avi-Aharon et al., 2020) and use the derivative of the logistic regression function, the Sigmoid function $\sigma(.)$, as a kernel $\mathcal{K}(.)$, that is for a variable $v$

$$\mathcal{K}(v) = \frac{d}{dv}\sigma(v) = \sigma(v)\sigma(-v) \,. \tag{12}$$

The integral of the function $\hat{f}(b, x, y)$ defined in Eq. (11) over the region gives the histogram value of the bin $b$ in a color channel $c$:

$$hist_c(b, R(x, y)) = \sum_{(x_n, y_n) \in R(x,y)} \left[ \sigma(\frac{I_c(x_n, y_n) - b - L/2}{B}) - \sigma(\frac{I_c(x_n, y_n) - b + L/2}{B}) \right] \,, \tag{13}$$

where $L = 1/256$ is the bin size, so that each bin represents a color value. We get the probability of each color value by dividing the sum of the histogram values by the size of the region $|R|$ and the number of channels $C$

$$p(I(x, y)) = p(b, R(x, y)) = \frac{1}{C \cdot |R|} \sum_{c \in \{r,g,b\}} hist_c(b, R(x, y)) \,. \tag{14}$$

The entropy of the pixel in the center of the patch is

$$E(I(x,y)) = - \sum_{(x_n,y_n) \in R(x,y)} p(I(x_n,y_n)) \log(p(I(x_n,y_n))) \tag{15}$$

$$= - \sum_{b \in [0,255]} p(b, R(x,y)) log(p(b, R(x,y))) . \tag{16}$$

The entropy module uses our entropy function, implemented as an autograd function in PyTorch, to realize the CUDA extension of the entropy computation. The input to the entropy function is the regions of the processed images $R(x,y)$. The CUDA extension allocates a GPU block for each region, hence, the grid size equals to the number of all possible regions for all images in the batch. The block size is 256 threads, i.e., a thread for each bin $b$. Each thread iterates over the whole region and computes the histogram of its corresponding bin value $b$ according to Eq. (13). Then, it normalizes the result by the region size and the number of channels to get the probability Eq. (14). And finally, each thread computes the entropy of the pixel Eq. (15), which is equivalent to the sum of the entropy of the histogram bins Eq. (16).[4]

## C  PROOFS

### C.1  PROOF OF THEOREM 1

We want to minimize the average probability of error of how well the masked image $I_t^M$ represents the information of the actual image $I_t$. First, we will work on a pixel level to bound the probability of error in the intensity of the $n^{\text{th}}$ pixel in location $x,y$ in images $I_t^M$ and $I_t$, denoted as $P_\varepsilon^{(n)} = \mathbb{P}(I_t(x,y) \neq I_t^M(x,y))$. Images can in general be considered lattices, with pixels being the random variables over intensities $\mathcal{B}$ (in our case these are the number of bins in the histogram as described in Appendix B).

Since the error event for the $n^{\text{th}}$ pixel is a binary event, it follows that $P_\varepsilon^{(n)}$ is a binary probability. Therefore, the average error probability over all pixels $N$ of the image can be computed as

$$\bar{P}_\epsilon = \frac{1}{N} \sum_n^N P_\varepsilon^{(n)} , \tag{17}$$

where $N = H \times W$ is the total number of pixels, computed as the product of the height $H$ and width $W$ of the image.

On a pixel-level, following Fano's inequality (Sabuncu, 2006; Scarlett & Cevher, 2019; Tandon et al., 2014) and assuming that the $n^{\text{th}}$ pixel in position $x,y$ in an image can take a value uniformly on $\mathcal{B}$, we get

$$E(I_t(x,y)|I_t^M(x,y)) \leq E_2(P_\varepsilon^{(n)}) + P_\varepsilon^{(n)} \log(|\mathcal{B}| - 1) , \tag{18}$$

where $E_2(\alpha) = \alpha \log \frac{1}{\alpha} + (1-\alpha) \log \frac{1}{1-\alpha}$ is the binary entropy function. Eq. (18) can be further bounded as $E_2(P_\varepsilon^{(n)}) \leq \log 2$, and $|\mathcal{B} - 1| \leq |\mathcal{B}|$. Moreover, since a pixel is uniform on $|\mathcal{B}|$ its entropy can be considered $E(I_t(x,y)) = \log |\mathcal{B}|$. Therefore, we can further bound Eq. (18) as

$$E(I_t(x,y)|I_t^M(x,y)) \leq \log 2 + P_\varepsilon^{(n)} \log(|\mathcal{B}|) \tag{19}$$

$$\Leftrightarrow E(I_t(x,y)|I_t^M(x,y)) - E(I_t(x,y)) \leq \log 2 + P_\varepsilon^{(n)} \log(|\mathcal{B}|) - E(I_t(x,y)) \tag{20}$$

$$\Rightarrow \mathcal{I}(I_t(x,y), I_t^M(x,y)) \geq (1 - P_\varepsilon^{(n)}) \log(|\mathcal{B}|) - \log 2 \tag{21}$$

$$\Leftrightarrow P_\varepsilon^{(n)} \geq 1 - \frac{\mathcal{I}(I_t(x,y), I_t^M(x,y)) + \log 2}{\log |\mathcal{B}|} . \tag{22}$$

The mutual information of two random variable is upper bounded by the minimum entropy of the marginals, therefore, $\mathcal{I}(I_t(x,y), I_t^M(x,y)) \leq \min(E(I_t(x,y)), E(I_t^M(x,y)))$. But the masked image will by definition represent less information than the original image, therefore,

---

[4]The entropy layer is part of the submitted code, that will be open-sourced upon acceptance.

we have $\mathcal{I}(I_t(x,y), I_t^M(x,y)) \leq E(I_t^M(x,y))$. We can take the *worse case scenario* and assume $\mathcal{I}(I_t(x,y), I_t^M(x,y)) \approx E(I_t^M(x,y))$ (Murphy, 2022). Therefore, Eq. (22) becomes

$$P_\varepsilon^{(n)} \geq 1 - \frac{E(I_t^M(x,y))) + \log 2}{\log |\mathcal{B}|} \, , \tag{23}$$

which bounds the error on the information carried by a pixel in the masked image.

To acquire the bound of the average error probability Eq. (17), we sum Eq. (23) for all pixels and divide by $N$ to get

$$\frac{1}{N}\sum_n^N P_\varepsilon^{(n)} \geq \frac{1}{N}\sum_n^N 1 - \frac{\sum_{x,y} E(I_t^M(x,y)) + \sum_n^N \log 2}{N \log |\mathcal{B}|} \tag{24}$$

$$\bar{P}_\varepsilon \geq 1 - \frac{\sum_n^N E(I_t^M(x_n, y_n))}{N \log |\mathcal{B}|} - \frac{\log 2}{\log |\mathcal{B}|} \, . \tag{25}$$

$\square$

## C.2 PROOF OF LEMMA 1

Let $X, Y$ be two discrete random variables, and the respective entropy is lower-bounded $E(X) \geq 0$, $E(Y) \geq 0$. The joint entropy between the two random variables can be expressed as

$$E(X,Y) = E(X) + E(Y|X) = E(Y) + E(X|Y) \, . \tag{26}$$

As any conditional entropy is greater or equal to zero, we get $E(X,Y) \geq E(X)$, and similarly $E(X,Y) \geq E(Y)$. If $E(X) \geq E(Y)$ then

$$E(X,Y) \geq E(X) \geq E(Y) \, . \tag{27}$$

If $E(Y) \geq E(X)$ then

$$E(X,Y) \geq E(Y) \geq E(X) \, . \tag{28}$$

Therefore, for discrete random variables, we get the bound of the joint entropy as

$$E(X,Y) \geq \max(E(X), E(Y)) \tag{29}$$

Following the previous derivation, when considering two images $I_1, I_2$ whose pixels are discrete random variables over intensities $\mathcal{B}$, and assuming that they are independent such that $E(I_1|I_2) = E(I_1)$, we can consider that the joint image entropy is approximated by the pixel-wise maximum of the two marginal entropies

$$E(I_1, I_2)(x,y) \approx \max(E(I_1(x,y)), E(I_2(x,y))) \, . \tag{30}$$

$\square$

Approximating the joint entropy by the lower bound gives a lower bound of the conditional entropy as well. Given that we maximize the masked conditional entropy in the CME loss, then maximizing the lower bound is an acceptable approximation. Meanwhile in the IT loss, using the lower bound of the joint entropy gives an upper bound for the mutual information, equivalent to its general upper bound of the minimum entropy of the marginals. Accordingly, maximizing the upper bound gives higher probability of good reconstruction, and the approximation of the joint entropy is acceptable.

## C.3 PROOF OF THEOREM 2

Due to the process of information transportation of the keypoints, we try to reconstruct the information each keypoint carries. Therefore, we can again leverage Fano's inequality (Scarlett & Cevher, 2019), to provide a lower bound for the average error probability of information transportation per keypoint.

We formalize our error probability of information transportation of the $i^{\text{th}}$ keypoint as the per-pixel error event $P_\varepsilon^{\text{IT}(i)} = \mathbb{P}(I_t(x,y) \neq R_t^{(i)}(x,y))$. Therefore, from Fano's inequality similarly to Eq. (18) we have for the per

$$E(I_t(x,y)|R_t^{(i)}(x,y)) \leq E_2(P_\varepsilon^{\text{IT}(i)}) + P_\varepsilon^{\text{IT}(i)} \log(|\mathcal{B}| - 1) \, , \tag{31}$$

And following similar manipulations as in Appendix C.1, we end up in the equivalent version of Eq. (22)

$$\mathcal{I}(I_t(x,y), R_t^{(i)}(x,y)) \geq (1 - P_\varepsilon^{\text{IT}(i)}) \log(|\mathcal{B}|) - \log 2 \Leftrightarrow \tag{32}$$

$$P_\varepsilon^{\text{IT}(i)} \geq 1 - \frac{\mathcal{I}(I_t(x,y), R_t^{(i)}(x,y)) + \log 2}{\log |\mathcal{B}|} \tag{33}$$

Similarly to Appendix C.1, we obtain the average error probability of information transportation of the $i^{\text{th}}$ keypoint by summing the per-pixel probabilities of Eq. (33) and averaging over number of pixels $N$

$$\bar{P}_\varepsilon^{\text{IT}(i)} \geq 1 - \frac{\sum_{(x,y)} \mathcal{I}(I_t(x,y), R_t^{(i)}(x,y))}{N \log |\mathcal{B}|} - \frac{\log 2}{\log |\mathcal{B}|} \tag{34}$$

Finally, considering all keypoint transportations as independent events, allows us to compute the joint average error probability for all keypoints as $\bar{P}_\varepsilon^{\text{IT (joint)}} = \sum_{i=1}^K \bar{P}_\varepsilon^{\text{IT}(i)}$, leading to the bound

$$\bar{P}_\varepsilon^{\text{IT (joint)}} \geq K - \frac{\sum_{i=1}^K \sum_{x,y} \mathcal{I}(I_t(x,y), R_t^{(i)}(x,y))}{N \log |\mathcal{B}|} - \frac{K \log 2}{\log \mathcal{B}}. \tag{35}$$

$\square$

## D    EVALUATION METRICS FOR OBJECT DETECTION AND TRACKING

Each keypoint should provide a representation of a feature in an object, and keypoints should be distinctive and distributed over the shallow scene. Keypoints assigned to empty spaces are considered unsuccessfully assigned. To judge the performance of our method, we developed metrics that use the object masks provided by CLEVRER (Yi et al., 2019) over a set of test videos $V$, each of which of length $T$.

### D.1    PERCENTAGE OF THE DETECTED OBJECT (DOP).

We consider an object detected if there is at least one keypoint on its mask $M_{obj}$. At each time frame, we count the percentage of detected objects with respect to the ground truth (GT) number of objects and average these values over the whole video. We get the final result by averaging the value over all the videos in the test dataset. Better detection corresponds to a higher percentage of detected objects.

$$M_{DOP} = \frac{1}{V \cdot T} \sum_{v=1}^V \sum_{t=1}^T \frac{N_{detected}}{N_{GT}}, \tag{36}$$

where $N_{detected}$ is the number of detected objects (at least one keypoint lies in the object mask)

$$N_{detected} = \sum_{obj \in O} \left[ \sum_{i=1}^K \mathbb{I}((x_i, y_i) \in M_{obj}) \right] > 0, \tag{37}$$

and $N_{GT}$ is the ground truth number of objects and $O$ is the set of all objects in the scene.

### D.2    PERCENTAGE OF TRACKED OBJECTS (TOP).

We consider an object tracked if there is at least one keypoint on its mask in the current and the previous timeframe. At each time frame, we count the percentage of tracked objects with respect to the ground truth (GT) number of objects and average these values over the whole video. We get the final result by averaging the value over all the videos in the test dataset. Better detection corresponds to a higher percentage of tracked objects.

$$M_{TOP} = \frac{1}{V \cdot T} \sum_{v=1}^V \sum_{t=1}^T \frac{N_{tracked}}{N_{GT}} \tag{38}$$

Where $N_{tracked}$ is the number of tracked objects (at least one keypoint lies in the object mask in time frames t and t-1)

$$N_{tracked} = \sum_{obj \in O} \left[ \sum_{i=1}^{K} [\mathbb{I}((x_i, y_i)_t \in M_{obj}^{(t)}] \cdot [\mathbb{I}((x_i, y_i)_{t-1} \in M_{obj}^{(t-1)})] \right] > 0. \tag{39}$$

### D.3 Unsuccessful keypoint assignment (UAK).

A keypoint is unsuccessfully assigned in a time frame if it does not belong to any object. We average the number of unsuccessful keypoint over the whole video, and then over test videos to get a global value over the testset

$$M_{UKA} = \frac{1}{V \cdot T} \sum_{v=1}^{V} \sum_{t=1}^{T} N_{uk} \, , \tag{40}$$

where $N_{uk}$ is the number of unsuccessful keypoints (does not belong to the sum of the masks)

$$N_{uk} = \sum_{i=1}^{K} \sim \mathbb{I}((x_i, y_i) \notin \sum_{obj \in O} M_{obj}) \, . \tag{41}$$

A lower unsuccessful keypoint assignment metric $M_{UKA}$ corresponds to better keypoints activation.

### D.4 Redundant keypoint assignment (RAK).

Assigning keypoints to areas already represented by other keypoints signals bad keypoint detection. The RAK metric accounts for the number of keypoints over the area of the object. The number of keypoints on an object mask should be proportional to its area $A_{obj}$. We assume a keypoint can represent some area of pixels $A_k$. If the keypoints cover the object, the RAK metric will have a value of 0, with higher values if more or fewer keypoints were assigned to that object.

$$M_{RAK} = \frac{1}{V \cdot T \cdot O} \sum_{v=1}^{V} \sum_{t=1}^{T} \sum_{obj \in O} \frac{|A_{obj} - A_k n_{obj}|}{A_{obj}} \, , \tag{42}$$

where $A_k$ is the representation area of a keypoint (e.g. average object areas in the dataset) $A_{obj}$ is the area of the object's mask and $n_{obj}$ is the number of keypoints assigned to the object

$$n_{obj} = \sum_{i=1}^{K} \mathbb{I}((x_i, y_i) \in M_{obj}) \qquad \text{for each } obj \in O \, . \tag{43}$$

The lower the value of $M_{RAK}$, the better, because more efficient, is the distribution of the keypoints.

The metrics collectively judge the efficacy of keypoint detection and tracking methods, where only detected objects can be tracked, so the DOP metric is an upper bound for the TOP metric. The value of the metric RAK will go to one in the case of not detecting any object, but can go higher in case of assigning redundandant keypoints to the same object. Following the previous observation, we recommend judging the the value of the RAK metric jointly with the value of the DOP metric.

## E Additional experimental analysis

### E.1 Ablation study

Our method for unsupervised keypoint discovery in video streams uses a collection of information-theoretic losses and some regularizers. In the ablation study, we investigate the role of each component and we discuss our design choices. In the following, we analyze different design choices like the entropy region size, the conditional entropy in the information transportation loss, and the regularizers.

**Ablation analysis** Using the proposed evaluation metrics, we analyzed several aspects of MINT on CLEVRER (Yi et al., 2019). We report the results in Table 5.

Since we compute the local entropy using the probability of the pixel value in its neighborhood region, we investigated the effect of the region size on the performance by varying the region size while using the information maximization (IM) (i.e., ME and MCE from Sec. 2.2.1) loss alone. The results show that a region of size $5 \times 5$ gives the highest values for the DOP and TOP metrics. We observe also that increasing the region size led to an increase in UAK, with a decrease in RAK; we hypothesize this is due to an over-smoothing effect of the bigger region, which leaks some information outside the objects. We noticed, on the other hand, an increase in the order of 30 minutes in the training time (50% of the training time) of one seed when increasing the region size by 2. Given the marginal improvement and the need for more resources, we adopted a region size of $3 \times 3$ for all of our experiments.

Then, we examined the information-theoretic losses without regularization, where we have an additional hyperparameter $\kappa$ to set the contribution of the conditional entropy in the information transportation (IT) loss. The results prove that adding conditional information improves the keypoint detection, with $\kappa = 0.5$ giving the best results for DOP and TOP followed by $\kappa = 0.9$. The value of RAK increases with lower $\kappa$, because keypoints seek the same areas of high information to reconstruct as much information as possible, leading to the redundant assignment. The introduction of conditional entropy in the IT loss, as describe in Sec. 2.2.2, helps mitigate this behavior by lowering the reconstruction error outside the transportation regions, i.e., the keypoint position in the current and the previous time frame. We highlight two values from this part; with $\kappa = 0.5$ we get the best scores for DOP and TOP, while $\kappa = 0.9$ trades off well all of the metrics (we call this model MINT w/o Reg. - highlighted in light blue, that is also referenced in Table 1).

Next, we investigate the regularization terms proposed in our method; (1) the movement loss controlled by the weight $m_d$ in the information transportation loss, (2) the overlapping loss (O), and (3) the active status loss (S). We experimented with all possible combinations of those regularizers. We can observe that the movement regularizer helps decrease the UAK metric, as this regularizer stabilizes the keypoint movement and constraints the keypoints from jumping into the background. The overlapping loss reduces the RAK value by almost half (from 3.982 to 2.079), but this comes with a higher UAK. The status loss pushes the UAK lower but comes at the cost of lower DOP and TOP. Introducing the overlapping and the status loss together allows better overall performance, where the overlapping loss pushes the DOP higher. We achieved the best trade-off across all metrics by setting $\kappa = 0.9$ while using all the regularizers (highlighted in light green). We adopt this option for our method MINT, and it proved to outperform the baselines both in the synthetic dataset (quantitatively proved in Table 1, and qualitatively shown in Fig. 4) and for realistic scenarios (Figs. 5 and 6).

Finally, we investigated the performance of the losses that work for single images, mainly the masked entropy loss (ME) with the regularizers: the active status loss (S) and the overlapping loss (O). This combination of losses does not use any temporal information, hence, it can operate on static images. We train this combination of losses on CLEVRER, operating on single images. We can observe that the model learned to track objects despite being trained on single images. However, we argue that, due to the structure of our training process using samples of a sequence of images, biased the model towards reducing the movement of the keypoints while attending to features, leading to good tracking performance. We call this ablation MINT w/o Temp., and we discuss it further later.

We show that if we have enough knowledge about the environment and we can decide on the suitable number of keypoint (e.g., K=10 keypoints for CLEVRER), then the IM loss alone is enough to get good performance (last row in Table 5), with low UAK and RAK, as the keypoint assignment is easier.

We further discuss the two major ablations MINT w/o Reg. and MINT w/o Temp. in detail regarding their performance qualitatively on the realistic datasets[5].

**MINT w/o Reg.** Our method MINT without the regularization terms, i.e., (1) removing the regularization for the keypoints' movement $m_d = 0$ in the information transportation loss, (2) removing the overlapping loss $\lambda_o = 0$, and (3) removing the active status loss $\lambda_s = 0$. Besides the quantitative results in Table 5, which show that the information-theoretic losses can detect and track objects better than the baselines (outperforming all of the baselines in the DOP and TOP metrics), we provide qualitative evidence of the performance of the proposed information-theoretic losses, where MINT w/o Reg. can detect and track the object in synthetic Fig. 12 and realistic scenes Figs. 10 and 11.

---

[5]Video results for the ablation study: `https://sites.google.com/view/mint-iclr/ablations`

Table 5: Ablation study on MINT losses. We report the statistics of the metric values over 5 seeds.
IM stands for the information maximization losses (ME + MCE), IT for information transportation, $\kappa$ decides the contribution of the conditional entropy in the IT loss, $m_d$ is the movement regularizer weight in the IT loss, O is the overlapping loss and S is the active status loss.
The ablations picked for MINT w/o reg. , MINT w/o Temp. and MINT are highlighted with light blue, light red, and light green consequently.
The weight scales used for all the ablations ($\lambda_{ME} = \lambda_{MCE} = 100$, $\lambda_{IT} = 20$, $\lambda_s = 10$, $\lambda_o = 30$, K=25).
The * near the method's name indicates a longer training time.

| Method | DOP ⇑ | TOP ⇑ | UAK ⇓ | RAK ⇓ |
|---|---|---|---|---|
| IM (3x3) | $0.951 \pm 0.042$ | $0.929 \pm 0.048$ | $\mathbf{6.777 \pm 1.369}$ | $3.885 \pm 1.090$ |
| IM (5x5)* | $\mathbf{0.956 \pm 0.036}$ | $\mathbf{0.932 \pm 0.043}$ | $8.276 \pm 1.428$ | $3.660 \pm 1.083$ |
| IM (7x7)* | $0.951 \pm 0.041$ | $0.926 \pm 0.048$ | $9.946 \pm 1.593$ | $\mathbf{3.098 \pm 0.959}$ |
| IM+IT ($m_d = 0, \kappa = 0$) | $0.917 \pm 0.072$ | $0.897 \pm 0.077$ | $\mathbf{3.543 \pm 1.529}$ | $5.096 \pm 1.587$ |
| IM+IT ($m_d = 0, \kappa = 0.5$) | $\mathbf{0.935 \pm 0.058}$ | $\mathbf{0.916 \pm 0.063}$ | $4.754 \pm 1.463$ | $3.982 \pm 1.226$ |
| IM+IT ($m_d = 0, \kappa = 0.9$) | $0.918 \pm 0.073$ | $0.897 \pm 0.078$ | $6.793 \pm 1.956$ | $2.478 \pm 0.865$ |
| IM+IT ($m_d = 0, \kappa = 1$) | $0.916 \pm 0.073$ | $0.895 \pm 0.078$ | $5.645 \pm 1.873$ | $\mathbf{2.336 \pm 0.768}$ |
| IM+IT ($m_d = 1$) | $0.883 \pm 0.097$ | $0.865 \pm 0.102$ | $1.665 \pm 0.954$ | $1.896 \pm 0.706$ |
| IM+IT ($m_d = 0$)+O | $\mathbf{0.921 \pm 0.066}$ | $\mathbf{0.898 \pm 0.073}$ | $7.769 \pm 1.880$ | $2.079 \pm 0.604$ |
| IM+IT ($m_d = 1$)+O | $0.879 \pm 0.102$ | $0.861 \pm 0.105$ | $2.196 \pm 1.228$ | $1.705 \pm 0.582$ |
| IM+IT ($m_d = 0$)+S | $0.851 \pm 0.114$ | $0.830 \pm 0.118$ | $1.057 \pm 0.666$ | $1.159 \pm 0.455$ |
| IM+IT ($m_d = 1$)+S | $0.842 \pm 0.116$ | $0.823 \pm 0.119$ | $1.060 \pm 0.735$ | $1.180 \pm 0.475$ |
| IM+IT ($m_d = 0$)+S+O | $0.859 \pm 0.112$ | $0.840 \pm 0.116$ | $1.130 \pm 0.710$ | $1.324 \pm 0.508$ |
| IM+IT ($m_d = 1, \kappa = 0.5$)+S+O | $0.844 \pm 0.120$ | $0.826 \pm 0.123$ | $1.100 \pm 0.716$ | $\mathbf{1.121 \pm 0.451}$ |
| IM+IT ($m_d = 1, \kappa = 0.9$)+S+O | $0.855 \pm 0.118$ | $0.838 \pm 0.121$ | $\mathbf{0.889 \pm 0.639}$ | $1.123 \pm 0.448$ |
| ME+S+O | $0.849 \pm 0.115$ | $0.826 \pm 0.119$ | $0.958 \pm 0.615$ | $1.142 \pm 0.446$ |
| MINT w/o Reg. | $\mathbf{0.918 \pm 0.073}$ | $\mathbf{0.897 \pm 0.078}$ | $6.793 \pm 1.956$ | $2.478 \pm 0.865$ |
| MINT w/o Temp. | $0.849 \pm 0.115$ | $0.826 \pm 0.119$ | $0.958 \pm 0.615$ | $1.142 \pm 0.446$ |
| MINT (ours) | $0.855 \pm 0.118$ | $0.838 \pm 0.121$ | $\mathbf{0.889 \pm 0.639}$ | $\mathbf{1.123 \pm 0.448}$ |
| K=10 | | | | |
| IM (3x3) | $0.879 \pm 0.085$ | $0.847 \pm 0.095$ | $1.662 \pm 0.781$ | $1.536 \pm 0.554$ |

On the other hand, the experiments justify the role of regularization in stabilizing keypoint detection and removing excessive keypoints. Fig. 12 from CLEVRER and Fig. 10 from SIMITATE shows a better distribution of keypoint when using MINT over MINT w/o Reg. Fig. 11 depicts the contribution of the regularization losses for economizing the number of used keypoints in the come-and-go situation, allowing MINT to outperform the other models.

**MINT w/o Temp.** our method MINT operating on single images without the losses that operate temporally over two images. MINT w/o Temp. requires (1) removing the masked conditional entropy loss $\lambda_{MCE} = 0$, and (2) the information transportation loss $\lambda_{IT} = 0$.
This ablation proposes good performance for keypoint detection on static images when using the losses that operate on a single image. Table 5 shows that MINT w/o Temp. can detect 85% of the objects in the scene while distributing the keypoint reasonably. Fig. 12 shows that the MINT w/o Temp. assigns keypoints to objects in the scene successfully. Fig. 10 shows that MINT w/o Temp. can detect the human and objects in the background, but due to the lack of temporal information, it doesn't concentrate on the moving objects (e.g. the hand of the human).

Overall, the full MINT model (cf. Table 5) trades off the need for good detection and tracking performance, but with a reasonable distribution of keypoints, to adequately represent the information in the video when minimizing our information theoretic losses (i.e., maximizing the covered information entropy spatiotemporally), as dictated by Theorems 1 and 2.

### E.2 HYPERPARAMETERS

Table 6 provides the hyperparameters used for CLEVRER (Yi et al., 2019) in our experiments. We use the same values for all other datasets, *i.e.*also for MIME (Sharma et al., 2018), SIMITATE (Memmesheimer et al., 2019), and MAGICAL (Toyer et al., 2020). The only exceptions are the activation threshold $\gamma$, the std for heatmap $\sigma_{G_i}$ and the threshold of the heatmap $\tau$ where we use

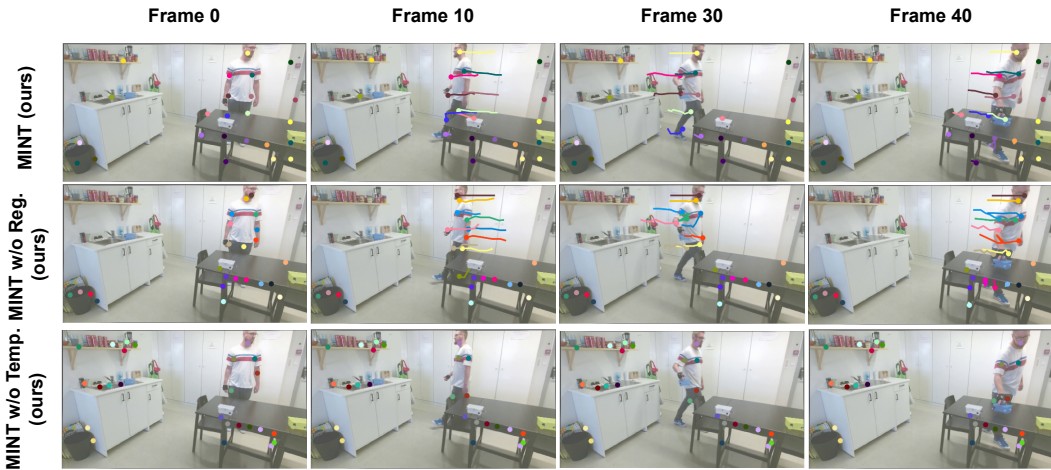

Figure 10: Crowded scenes. A video from SIMITATE dataset with a human moving in a room. We compare MINT with its ablations MINT w/o Reg. and MINT w/o Temp. from Appendix E.1

Table 6: Hyperparameters

| Parameter name | value | Parameter name | value | Parameter name | value |
|---|---|---|---|---|---|
| learning rate | 0.001 | clip value | 10.0 | weight decay | 0.00001 |
| epochs | 100 | num keypoints $K$ | 25 | number of stacked frames | 3 |
| activation threshold $\gamma$ | 15 | entropy region size $\sqrt{|R|}$ | 3 | std for heatmap $\sigma_{G_i}$ | 9.0 |
| Threshold for heatmap $\tau$ | 0.1 | Thresholded heatmap scale $\eta$ | 3.5 | CE contribution (IT) $\kappa$ | 0.5 |
| movement weight (IT) $m_d$ | 1.0 | ME weight $\lambda_{\text{ME}}$ | 100 | MCE weight $\lambda_{\text{MCE}}$ | 100 |
| IT weight $\lambda_{\text{IT}}$ | 20 | active status weight $\lambda_{\text{s}}$ | 10 | overlapping weight $\lambda_{\text{o}}$ | 30 |

values depending on the size of the input image (*i.e.*, $\gamma = 15$, $\sigma_{G_i} = 9.0$, $\tau = 0.1$ for MIME, $\gamma = 10$, $\sigma_{G_i} = 9.0$, $\tau = 0.5$ for SIMITATE, $\gamma = 10$, $\sigma_{G_i} = 7.0$, $\tau = 0.3$ for MAGICAL). Our method requires a sequence of 2 frames for the loss computation, and we found that the batch size does not affect the training and can be chosen based on the available GPU resources.

### E.3 BASELINES

**Video Structure** (Minderer et al., 2019) is an unsupervised method for learning keypoint-based representation from videos. Video structure learn a keypoints detector $\phi^{det}(v_t) = x_t$ for a video sequence $v_t$ that captures the spatial structure of the objects in each frame in a set of keypoints $x_i$. Additionally, it learns a reconstruction model $\phi^{rec}$ that reconstructs frame $v_t$ from its keypoint representation $x_t$ and the first frame of the sequence $v_1$. An additional skip connection from the first frame to the reconstruction model output changes its actual task to predict $v_t - v_1$; hence $v_t - v_1 = \phi^{rec}(v_1, x_t)$.

The keypoint detector is trained to optimize three losses:

(1) L2 image reconstruction loss

$$\mathcal{L}_{\text{image}} = \sum_t ||v - \hat{v}||_2^2 \tag{44}$$

where $v$ is the true and $\hat{v}$ is the reconstructed image.

(2) Temporal separation loss penalizes the overlap between trajectories within a Gaussian radius $\sigma_{\text{sep}}$

$$\mathcal{L}_{\text{sep}} = \sum_k \sum_{k'} exp(-\frac{d_{kk'}}{\sigma_{\text{sep}}}) \tag{45}$$

where $d_{kk'} = \frac{1}{T}\sum_t ||(x_{t,k} - \overline{x}_k) - (x_{t,k'} - \overline{x}_{k'})||_2^2$ is the distance between the trajectories of keypoints $k$ and $k'$.

(3) Sparsity loss adds an L1 penalty on the keypoint intensity $\mu$ (the mean value of the corresponding

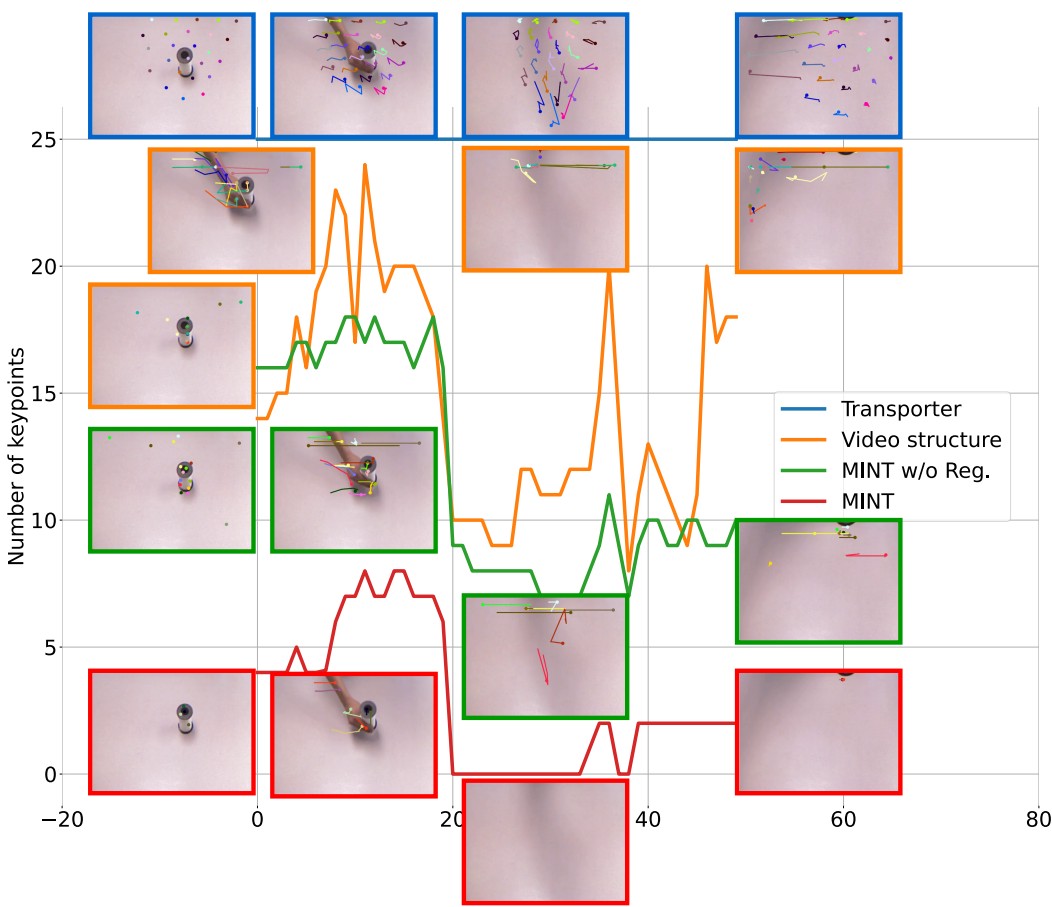

Figure 11: Come-and-go scenario. In a manipulation video from MIME, the hand enters after the start of the video and departs before the end. We plot the number of active keypoints w.r.t. timesteps. The results show the role of regularization in improving resource assignment and economizing the number of active keypoints.

feature map) to encourage keypoints to be sparsely active

$$\mathcal{L}_{\text{sparse}} = \sum_k |\mu_k| \tag{46}$$

a keypoint is active if the intensity is higher than a specific threshold $th_\mu$, the threshold is a hyperparameter that has to be tuned depending on the video.

**Transporter** (Kulkarni et al., 2019) is a neural network architecture for discovering keypoint representations in an unsupervised manner by transporting learned image features between video frames using the keypoint bottleneck. During training, spatial feature maps $\phi(x)$ and keypoint coordinates $\psi(x)$ are predicted for a source frame $x_s$ and a target frame $x_t$ using a ConvNet and KeyNet (Barroso-Laguna et al., 2019). The keypoint coordinates are transformed into Gaussian heatmaps $h_{\psi(x)}$.

A trasnported features map $\hat{\phi}(x_s, s_t)$ is generated by suppressing both sets of keypoint location in $\phi(x_s)$ and composing into the feature maps around the keypoints from $x_t$:

$$\hat{\phi}(x_s, s_t) \triangleq (1 - h_{\psi(x)_t}) \cdot (1 - h_{\psi(x)_t}) \cdot \phi(x_s) + h_{\psi(x_t)} \cdot \phi(x_t) . \tag{47}$$

An additional refiner net learns to map the transported features maps into an image $\hat{x}_t$. The learning objective is reconstructing the target image $x_t$ from the process. Hence, the Transporter optimizes the L2 reconstruction error $\mathcal{L} = ||x_t - \hat{x}_t||_2^2$

**Transporter-modified** is a modified version of the transporter baseline (Kulkarni et al., 2019). The original implementation of the method has two potential bottlenecks; (1) the feature maps $\phi(s)$ have a receptive field of size 24 for each position, for an input of size 128x128; (2) the resolution of the

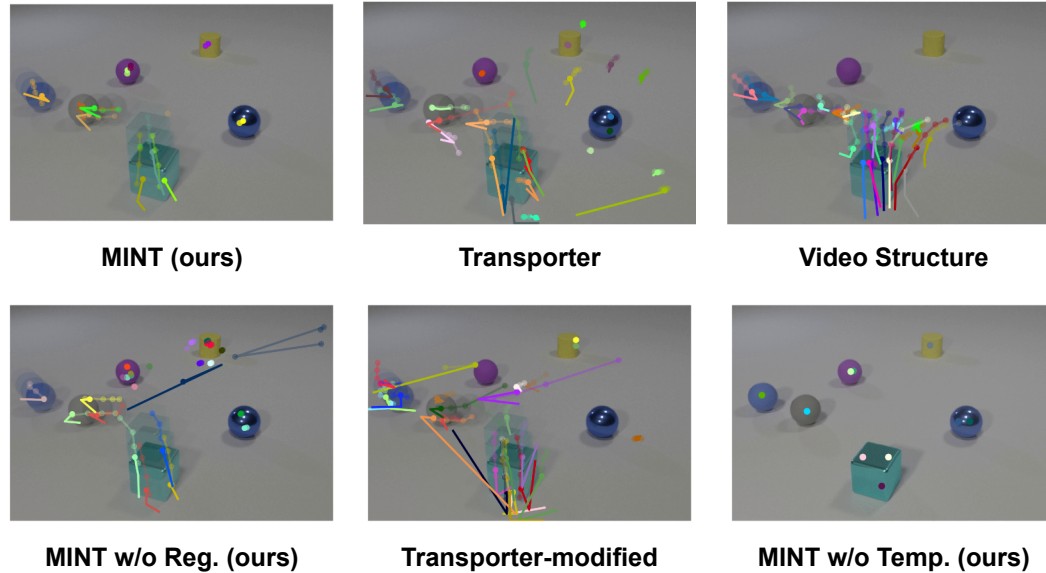

Figure 12: Qualitative results on CLEVRER dataset for Task I (object detection and tracking) and Task II (learning dynamics). We include additional results for all baselines Appendix E.3.

feature maps between which the features are transported is 32x32. For fair comparison to our method, which uses an entropy region of size 3x3, we modified the network architecture of ConvNet $\phi(x)$ to have (1) a receptive field of 7 and (2) a feature map of size 122x122. We call the new architecture Transporter-modified.

The experimental results show that the Transporter-modified model outperforms the original Transporter in the quantitative evaluation on CLEVRER dataset (Yi et al., 2019) (cf. Table 1). An interesting observation from visual results in Fig. 12, one can see keypoints of the original Trasnporter (top middle image) are more stable than those of the Transporter-modified (bottom middle image). We argue this behavior is due to the smaller receptive field leading the model to assign keypoints to features instead of objects, and thus keypoints jump to similar features in different objects.

### E.4    INTERACTION NETWORK ARCHITECTURE

The interaction network (Battaglia et al., 2016) is a model developed for learning the interaction relations between physical objects to infer the physics of the environment. The interaction network treats the objects as nodes of a graph, with the relations as edges. In our case, we use the keypoints as object nodes, with the coordinates, status, and positional encoding as features. We form a fully connected graph of the keypoints, with no relational features for edges.

The interaction network used in our experiments has two sub-models; a relational model and an object model. The relational model uses the relational information and object attributes to predict the effects of all interactions. The object model uses the effects to update the features of the object. We encode node features before passing them to the interaction network. After one pass through the interaction network, we decode the features into coordinates for the prediction task, and we add another prediction head for the action decoding in the imitation learning task.

### E.5    IMITATION LEARNING RESULTS

**CNN-agent:** The CNN agent is trained from scratch for every environment. Note that the state space for the CNN agent, the image pixels, is one order of magnitude higher than the keypoints' features. For fair comparison, we train the CNN agent longer (twice the epochs used for training the MINT-based agent to counteract for MINT's pretraining).

The CNN feature extractor consists of 5 convolution blocks, each consisting of a 2D convolutional layer with a ReLU activation function and a batch normalization layer. The input to the model is a

**MoveToRegion**

**MoveToCorner**

**MakeLine**

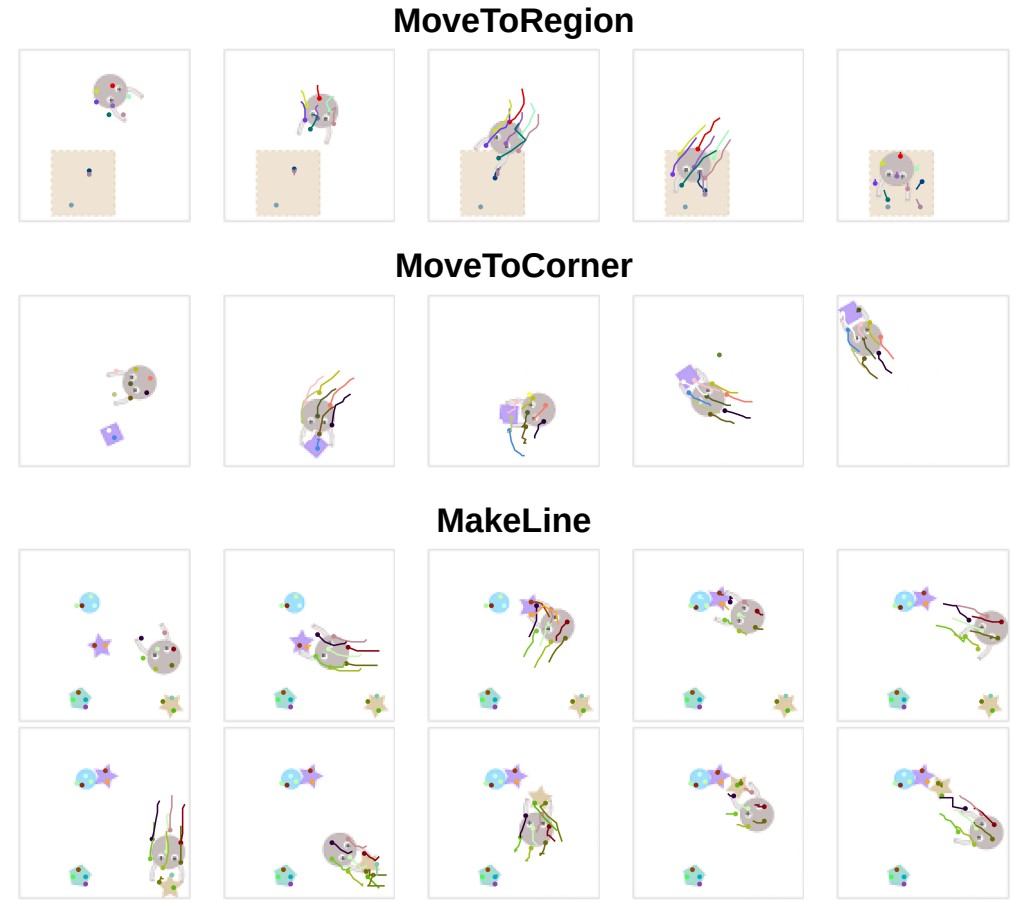

Figure 13: Rollouts from MINT agent in MAGICAL (Toyer et al., 2020) dataset.

sequence of 4 color images stacked over the channel axis; hence, the input size is $12 \times 96 \times 96$. The layers have 64, 128, 128, 128, and 128 filters, with a kernel size of 3 and stride of 2, except the initial layer, which has a kernel size of 5 and stride of 1. The output of the last block is flattened and passed to a linear layer to provide the final features. A policy model uses the feature to infer the actions. The policy model is a multi-layer perceptron with 4 linear layers of sizes 128, 64, 32, and 32. The output matches the action dimension of MAGICAL environment which is 18.

We, also, provide visualizations of the learned policies of the MINT-based agent on the three environments of MAGICAL in Fig. 13. The visualization shows that MINT can assign reasonable keypoints for the agent and all the objects in the environment. The imitation agent can solve the first two tasks **MoveToRegion** and **MoveToCorner**, but it struggles with the last task **MakeLine**. The agent receives a score of 1.0 when it sorts all 4 objects in one line, while it gets a score of 0.5 for putting 3 out of 4 in one line. Our imitation agent could (occasionally) sort only 3 out of 4 in the depicted environment (which led to an overall 0.2 mean score over 5 seeds – Table 3), despite being able to assign keypoints to all objects. The results suggest that there is a problem in encoding the relational features between keypoints, hindering the agent to reason upon getting the right locations. We argue that further investigation of the appropriate model to pool information from the keypoints is necessary to solve this harder task, but this is out of scope of the current work.

### E.6    ADDITIONAL VIDEO RESULTS

We provide additional video results both as supplemental material and on the website of our project: https://sites.google.com/view/mint-iclr

## F  CODE

We include our code in the supplemental material, and we plan to publish our code under an open-source license to accompany the paper upon acceptance. We provide instructions to run the code, with sample datasets to reproduce the results in the paper.

The implementation of video structure (Minderer et al., 2019) [6] uses outdated libraries. Due to compatibility reasons, we reimplemented their code in PyTorch with our best effort.

We adapted the implementation of Transporter from Li et al. (2020) [7] into the codebase of MINT [8].

## G  EXTENDED RELATED WORK DISCUSSION

**Image information entropy.** Our work draws inspiration from classical approaches in saliency detection in images that use local information to detect salient objects (Kadir & Brady, 2001; Bruce & Tsotsos, 2005; Fritz et al., 2004; Renninger et al., 2004; Borji & Itti, 2012). We use the entropy as a measure of saliency. Bruce & Tsotsos (2005) proposes that regions with high self-information typically correspond to salient objects, and Alexe et al. (2010) quantified objectiveness by self-information approximated via center-surround feature differences. Gopalakrishnan et al. (2020) used local predictability as measure of saliency to learn keypoints.
However, image information entropy has applications in different computer vision problems, apart for saliency detection. Sabuncu (2006) uses entropy and other information-theoretic measures for image registration. Ferraro et al. (2002) propose an entropy-based representation of image information with application to pattern recognition and active vision. Other applications are medical image analysis (Hržić et al., 2019), nuclear detection (Hamahashi et al., 2005), image compression (Minnen et al., 2018), and measuring image randomness (Wu et al., 2013). Our method proposes information-theoretic losses based on the local entropy as a measure of information.

**Representation learning.** The idea of extracting sparse feature representations of high-dimensional visual data is dominant in computer vision and machine learning research (Harris et al., 1988; Lowe, 2004), and connects to the functioning of the human visual system (Marr, 2010). Such sparse representations are generally known as PoI, which are 2D locations that are stable and repeatable under various lighting conditions and viewpoints (DeTone et al., 2018). Traditional geometric computer vision methods relied on the extraction of hand-crafted feature descriptors (Lowe, 2004; Rublee et al., 2011) for tasks like localization (Schmid et al., 2000; Mur-Artal et al., 2015). In the deep learning era, CNN architectures have proven superior to handcrafted features (Yi et al., 2016; DeTone et al., 2018; Song et al., 2020; Zheng et al., 2017). Deep approaches extract clouds of PoI that are useful for correspondence searching in visual place recognition from different viewpoints (Hausler et al., 2021), or pose-estimation for control (Florence et al., 2019).
Related to our method are object-centric approaches Singh et al. (2021); Locatello et al. (2020); Dittadi et al. (2021), which aim to learn abstract representations for objects in a scene. Our approach to keypoint discovery, alongside our metrics, are directed to learning and evaluating keypoints as an object-centric representation. In that direction, (Hung et al., 2019; Zhang et al., 2018a) finetune pre-trained ImageNet features such that each filter represent an object part. SCOPS (Hung et al., 2019) uses a way of computing saliency based on local connextivity of superpixels, as defined in (Zhu et al., 2014), and poses a saliency constraint in part discovery. Differently from SCOPS, we use local image entropy to quantify saliency, and provide method that can not only detect important parts in static frames, but can also represent temporal change in information, due to changes in a video. ICNN (Zhang et al., 2018a) optimizes mutual information and spatial entropy for interpretability. In particular, the authors try to enforce part template matching to a single target object part, that should be interpretable.
Differently to these methods, we can represent scenes with multiple objects in videos, and not in static images with single objects, in which the information dynamically changes, optimizing for information coverage and temporal consistency. Moreover, our method learns from scratch and uses feature maps to locate keypoints representing part features. The keypoints in our approach are information transporters of their local information (in the heatmap around them), while we can

---

[6] https://github.com/google-research/google-research/tree/master/video_structure
[7] https://github.com/pairlab/v-cdn
[8] The baselines are part of the submitted code, that will be open-sourced upon acceptance.

economize the use of keypoints to adequately represent the video information, with minimum number of keypoints.

**Keypoint learning.** Keypoints represent a class of PoI that have a semantic entity, e.g., representing objects (Duan et al., 2019), or human joints (Cao et al., 2017; McNally et al., 2022), but most methods rely on explicit annotations of keypoint locations. Related to our work are methods that rely on image reconstruction for detecting keypoints in video streams in an unsupervised fashion (Jakab et al., 2018; Zhang et al., 2018b; Minderer et al., 2019; Kulkarni et al., 2019; Gopalakrishnan et al., 2020). Jakab et al. (2018) use an autoencoder architecture with a differentiable keypoint bottleneck trained on the difference between a source and a target image, trying to restrict the information flow. MINT also uses a differentiable keypoint representation, but it operates on the output of an hourglass architecture. Our results suggest that learning to redistribute the information after compression is beneficial for keypoint discovery (Newell et al., 2016). Minderer et al. (2019) use a similar architecture as Jakab et al. (2018) but operate on video sequences for detecting keypoints, using the intensity of the bottleneck heatmap as an indicator of the importance of a keypoint. Setting up a threshold on the intensity is challenging and domain-specific. Contrarily, we learn a binary classification of active/inactive keypoints and optimize the number of keypoints used in every frame, as we showed in our results. Kulkarni et al. (2019) propose feature transportation in the keypoint bottleneck of Jakab et al. (2018) before reconstruction. Inspired by this idea, MINT performs information transportation and waives the need for image reconstruction, which would require an additional appearance encoder and a reconstruction decoder from keypoints. Gopalakrishnan et al. (2020) devised a three-stage architecture that first learns a spatial feature embedding, then solves a local spatial prediction task related to object permanence, and finally converts error maps into keypoints. While this method employs local information, it trains three architectural modules separately, unlike (Kulkarni et al., 2019; Minderer et al., 2019) and MINT that use a single end-to-end pipeline for keypoint discovery.

**Information-theoretic approaches in Machine Learning.** Information-theoretic principles proved advantageous in training and understanding machine learning models (Yu et al., 2021). Different information measures aim to describe a random variable's behavior due to a probability density function. The probability density function is normally unknown, and machine learning methods usually estimate it (Pardo, 2018; Avi-Aharon et al., 2020).In our method, we use kernel density estimation (KDE) (Parzen, 1962) to estimate the probability density function for a region of pixels. Various information-theoretic quantities were used in machine learning for different applications; examples are the cross-entropy loss for classification (Good, 1992; Goodfellow et al., 2016), maximum entropy regularization in reinforcement learning (Peters et al., 2010; Haarnoja et al., 2018), mutual information for self-supervised learning and interpretability (Rakelly et al., 2021; Zhang et al., 2018a), and KL divergence for training deep energy models (Yu et al., 2020). Our approach uses Shannon's definition of entropy (Shannon, 2001) to compute the local image entropy. With image entropy, we estimate joint entropy, conditional entropy, and mutual information and develop our information-theoretic losses.

The information bottleneck is an information-theoretic framework that analyzes learning dynamics in deep neural networks (Tishby et al., 2000). The information maximization principle (InfoMax) (Linsker, 1988) treats the neural network as an information channel and aims to maximize the information transferred through the network. Recent methods in computer vision and natural language processing use the InfoMax principle for self-supervised learning (Oord et al., 2018; Hjelm et al., 2018; Kong et al., 2019). Our method adopts the treatment of the neural network as an information channel in the information maximization loss and extends it to treat the keypoints as transmitters of information, while being completely unsupervised.

