# OpenReview forum: "An information-theoretic approach to unsupervised keypoint representation learning"
_ICLR.cc/2023/Conference — Submitted to ICLR 2023_

### Official Review · Reviewer_PH7P · 2022-10-20

**Confidence:** 4
**Correctness:** 3
**Technical Novelty And Significance:** 3
**Empirical Novelty And Significance:** 3
**Recommendation:** 6

**Clarity, Quality, Novelty And Reproducibility:**

Overall, this paper is well-written except the section 2.2.2 for me. The quality of this paper is also good, but if more analyses, such as the ablation studies for the losses, were reported, it should be much better. The novelty is good enough when seeing their manuscripts and experimental results for transportation and sparsity on the usage of keypoints. They mention that they will share their code upon acceptance.

**Strength And Weaknesses:**

Strength
- Well-designed model and losses. I could agree with why they propose and applies the losses.
- Good visualization, especially for transportation and sparse keypoints usage in Figures 5 and 6.
- Investigating Imitation learning. Usually, keypoint representation learning or similar representation learning papers, such as Object-Centric Learning, validate their method for detection, segmentation, or prediction [3,4,5]. Still, they evaluate their approach, not just detection and prediction, but also imitation learning.
- Mentioning their limitations because their model uses local image entropy.

[3] Locatello, Francesco, et al. "Object-centric learning with slot attention." Advances in Neural Information Processing Systems 33 (2020): 11525-11538.

[4] Dittadi, Andrea, et al. "Generalization and robustness implications in object-centric learning." arXiv preprint arXiv:2107.00637 (2021).

[5] Singh, Gautam, Fei Deng, and Sungjin Ahn. "Illiterate dall-e learns to compose." International Conference on Learning Representations. 2021.

Weaknesses
- Overall, the paper is well-written and easy to follow, but section 2.2.2 is not easy, at least for me.
    - In Figure 3, you mentioned The target information $E(T_t)$ is the sum of the conditional entropy E(I_t|I_{t-1}) and the patch around the keypoint at time $t-1$, but in paragraph, $E(T_t)$ is described differently. It looks like the summation of the patch around the keypoint at time $t$ and the new information not represented through the keypoint at time $t$.
    - In Figure 3, $E(S_t)$ looks like many regions with high entropy are not represented through keypoints. What is the original scene? From the entropy figure, it is not easy to understand.
    - It is hard for me to follow that "In a temporal sequence of frames, optimized keypoint information transport means that we can reconstruct the information of the current frame $E(I_t)$ using information from previous frames $E(I_{t−1})$ and the keypoint information from both frames."
    - You mentioned being inspired by [1]. When comparing with the loss in [1], you apply the new term, $E(I_t | I_{t-1}) \bigodot (1-h_i^{(t)})$. It can be analyzed like the new information (dynamics) but not represented through keypoints at time $t$. Why must it be applied to be the target information?
- They propose a set of multiple losses, but they didn't do the ablation studies on what happens if one or some of the losses are not used. It can be interesting and show the role of each loss empirically, so I hope to be updated in the revised version.
- The main baselines are Transporter [1] and Video Structure [2]. Could you make an additional section to discuss the difference in detail?

**Summary Of The Paper:**

This paper proposes a new keypoint representation learning method through novel losses inspired by information theory. One of the losses encourages the keypoints to represent the scene without information loss, and the other guides the keypoints to represent well the dynamics of the video through conditional entropy and the keypoints in the previous frame. In addition, they proposed overlapping loss to prevent the multiple keypoints from representing near regions and active status loss to represent the scene with sparse keypoints.

They empirically show their method outperforms baselines (Transporter [1] and Video Structure [2]). In particular, they show their method can represent the scene with a smaller number of keypoints and better transportation (e.g., when some of the objects in the scene move, only the relevant keypoints track).

[1] Tejas D Kulkarni, Ankush Gupta, Catalin Ionescu, Sebastian Borgeaud, Malcolm Reynolds, Andrew Zisserman, and Volodymyr Mnih. Unsupervised learning of object keypoints for perception and control. Advances in Neural Information Processing Systems, 32, 2019.

[2] Matthias Minderer, Chen Sun, Ruben Villegas, Forrester Cole, Kevin P Murphy, and Honglak Lee. Unsupervised learning of object structure and dynamics from videos. Advances in Neural Information Processing Systems, 32, 2019.

**Summary Of The Review:**

This paper proposes a new method for keypoint representation learning inspired by information theory. Through the proposed objective, they show better detection and prediction performance. Moreover, they also show the sparsity of using the keypoint representation and more stable transportation on the video dataset. However, at least for me, the explanation for the transportation loss is not clear, and the absence of the ablation studies of each loss makes me less understand their method.

---

> ### Author Response · Authors · 2022-11-19
> **Response to Reviewer PH7P**
>
> We thank the reviewer for the time and effort invested in providing constructive feedback. We are glad that the reviewer recognizes the novelty and contributions of our work. We also thank the reviewer for pointing out the variability of our downstream tasks compared to other representation learning methods, such as the ones mentioned by the reviewer [3-5], which we now include and discuss in the extended related work section in Appendix G - page 30.
>
> > - Overall, the paper is well-written and easy to follow, but section 2.2.2 is not easy, at least for me.
>     - In Figure 3, you mentioned The target information $E(T_t)$ is the sum of the conditional entropy $E(I_t|I_{t-1})$ and the patch around the keypoint at time t−1, but in the paragraph, $E(T_t)$ is described differently. It looks like the summation of the patch around the keypoint at time t and the new information not represented through the keypoint at time t.
> >    - In Figure 3, $E(S_t)$ looks like many regions with high entropy are not represented through keypoints. What is the original scene? From the entropy figure, it is not easy to understand.
> >    - It is hard for me to follow that "In a temporal sequence of frames, optimized keypoint information transport means that we can reconstruct the information of the current frame $E(I_t)$ using information from previous frames $E(I_{t−1})$and the keypoint information from both frames."
> >    - You mentioned being inspired by [1]. When comparing with the loss in [1], you apply the new term, $E(I_t|I_{t−1})\odot (1−h_i^{(t)})$. It can be analyzed like the new information (dynamics) but not represented through keypoints at time t. Why must it be applied to be the target information?
>
> We thank the reviewer for pointing out some inconsistencies in the explanation of the transportation loss. We updated section 2.2.2 and the caption of Fig. 3 to highlight the operations better. In particular, Figure 3 represents the transportation operation for a single keypoint; it shows the regions with high entropy that are not covered by that keypoint. After performing the information transportation (IT) operation for a keypoint, we get the reconstructed information image which consists of the source and the target. We have two sources of information mismatch between the reconstructed information and the entropy of the current image; (1) transportation error: the error inside the regions covered by the keypoint in the current and previous timestep means bad transportation, and (2) pixel changes outside of the regions, which different keypoints may cover. We use conditional entropy to mitigate the second error, as it helps to concentrate more on the keypoint transportation than reconstructing information. Dropping the conditional entropy may lead the keypoint to seek regions of high entropy trying to reduce the second source of error instead of attending to the same information pattern. This is also discussed in Appendix E.1, page 24, in the discussion of the ablation of the conditional entropy contribution in the information transportation (IT) loss, by controlling the $\kappa$ hyperparameter (results in Table 5, page 25). We note, "The results prove that adding conditional information improves the keypoint detection, with κ = 0.5 giving the best results ... helps mitigate this behavior by lowering the reconstruction error outside the transportation regions, i.e., the keypoint position in the current and the previous time frame."
> We would appreciate further feedback from the reviewer to solve any presentation issue in that section.
>
> >They propose a set of multiple losses, but they didn't do the ablation studies on what happens if one or some of the losses are not used. It can be interesting and show the role of each loss empirically, so I hope to be updated in the revised version.
>
> We thank the reviewer for this comment. We conducted extended ablations reported in section E.1 of the appendix, and we added the important ablation MINT w/o Reg. in the experimental section of the main paper. We discuss in detail the quantitative and qualitative results of the ablations (Figures 10, 11, 12) in Appendix E.1 - page 23.
>
> >The main baselines are Transporter [1] and Video Structure [2]. Could you make an additional section to discuss the difference in detail?
>
> We added a discussion on the baselines in Appendix E.3 - page 26. We hope this presentation clarifies the fundamental differences of the baselines w.r.t. our method.
>
> >They mention that they will share their code upon acceptance.
>
> We want to point out that our code was part of the supplementary material in the original submission, and we included it again in the resubmission. The code contains instructions on how to set up the environments and reproduce the results. We will indeed make the code available in a public GitHub upon acceptance.
>
> We hope our updates are well-aligned with the requests of the reviewer. In case there are further comments, please let us know.

---

### Official Review · Reviewer_Nich · 2022-10-23

**Confidence:** 4
**Correctness:** 3
**Technical Novelty And Significance:** 3
**Empirical Novelty And Significance:** 3
**Recommendation:** 6

**Clarity, Quality, Novelty And Reproducibility:**

- The paper is clear and well-written.
- To the best of reviewer’s knowledge, the proposed method and the introduced evaluation metric are novel.
- For reproducibility, the authors provide some of the hyper-parameters for the proposed model, but it is not sufficient to reproduce the experiment result. It is also not clear how to implement the parallelized entropy layer, it would be nice if the authors could release the code for future research.

For Clarity

- One small thing: It would be nice to provide a theoretical derivation on how to approximate a joint pixel-wise entropy as the max over the two marginals, i.e., how one can have $E\left(I_t, I_{t-1}\right)=\max \left(E\left(I_t\right), E\left(I_{t-1}\right)\right)$

**Strength And Weaknesses:**

Strengths:

- The research topic of unsupervised keypoint detection is of significance to the community.
- The method is original, and the evaluation results do demonstrate a consistent improvement over baselines on several tasks.
- The proposed parallel entropy layer is novel and could be useful for future works.
- The evaluation is well-designed. It includes a range of tasks including object detection and tracking, dynamic prediction, real-world object discovery, and imitation learning, which thoroughly demonstrate the quality of learned keypoints.
- The authors also introduce a new set of metrics to evaluate object detection and tracking quality, which can be considered one contribution.
- The additional video material provided in the anonymous link and supplementation are much appreciated, they provided a clear insight into the benefits of the method.

Weakness:

The main weakness of the paper is the lack of some key ablation studies, which make the claim that the method represents the scene better than the baselines less convincing.

- In addition to using local pixel entropy as input, the method also introduces several regularization terms in the training objective. It is possible that the improved performance comes from these new loss terms which are not included in the baseline models, they include:
    1. Minimizing the norm of the distance traveled by each keypoint in the IT loss to regularizing excessive keypoints’ movement
    2. The overlapping loss that encourages the keypoint to spread over the image
    3. The additional activation status variable, and the corresponding active status loss that control (reduce if possible) the number of active keypoint

    These additional terms could be considered contributions of the paper, but they should be separated from the main claim that the information-based method is better than the CNN-based method. Therefore, two sets of experiments can be considered in the paper, (1) the ablation on the proposed method to show the effect of each regularization term, or (2) incorporating these terms in the baseline models when comparing them with the proposed method.

- The qualitative results show a clear advantage of the proposed method which is very impressive. That is, for the proposed method, the movement of an object only affects the keypoints attached to its body, but for the baseline models, it also causes the movement of many keypoints surrounding the object. However, it is unclear if this behavior is due to architectural design, namely the receptive filed and transporting resolution.
    1. Receptive filed. In the baseline model, e.g., transporter, the CNN features that the model learns to transport has a receptive field of 24 (input size is 64x64) for each position (computed based on the implementation the paper use [1]), so each pixel contains the information of a large area of the image. This might explain why the surrounding keypoints will also follow the movement of the object - they capture part of the object’s information. On the other hand, the image entropy is designed to represent local information, the entropy region size is three. Is it possible that the baselines can enjoy similar improvement with a careful receptive field design?
    2. Resolution. Based on transporter implementation [1], the resolution of the feature map to transport is 16x16, and the input image resolution is 64x64, both are smaller than the counterpart of the proposed method and could inevitably cause loss of information. Please provide some investigation into this factor.

[1] [https://github.com/pairlab/v-cdn](https://github.com/pairlab/v-cdn)

**Summary Of The Paper:**

This work proposes an information-based unsupervised keypoint discovery method. Unlike other works that transport the CNN features between image frames, this work proposes to transport the information represented by the entropy of pixel color computed in a local region, and the authors argue that it provides a better inductive bias for unsupervised keypoint detection. To verify the hypothesis, they provide a quantitative and qualitative evaluation of synthetic and real-world datasets and demonstrate improvements over baselines on all tasks.

**Summary Of The Review:**

The problem this paper is addressing is of significance to the community. The proposed method is novel, and the evaluation results also show improvement over the baselines. The main concern is the lack of certain key experiments and ablation studies that are required to justify the claim and the advantage over existing works. The score could be increased if the above concerns are properly addressed.

---
## Post-rebuttal updates (edited):

I'm grateful to the authors for the comprehensive feedback and extra experiments in the revision.

1. The extra experiments on the additional loss terms for keypoint detection and tracking reveal that the information contributes significantly to the performance. In fact, in some cases, the model performs better without the additional loss terms, which is interesting. It would be nice to include this setting in other metrics and experiments as well.
2. I also appreciate the additional findings on Transporter-modified. The results indicate that Transporter performance could be enhanced by fine-tuning its architecture, which seems to imply that a deeper investigation into the model design is required to reveal the cause of the performance difference.

Additionally, after reviewing the other reviewers' comments, I concur with Reviewer NNTd that the paper is theoretically overstated. For example, it is unclear to me how the proof in Appendix C.2 can justify the approximation $E\left(I_t, I_{t-1}\right)=\max \left(E\left(I_t\right), E\left(I_{t-1}\right)\right)$. Neither the assumption that two consecutive frames are independent nor the statement that the joint will be equal to the max of two marginal are theoretically sound. It appears to be an empirical discovery rather than a result of mathematical analysis.

Despite the writing being theoretical sugar coating, the performance difference between the model and baselines demonstrated in the experiments and the website videos still clearly indicate that this is a promising approach in unsupervised keypoint detection. Therefore, I am leaning towards acceptance and have increased the score to 6.

---

> ### Author Response · Authors · 2022-11-19
> **Response to Reviewer Nich (1/2)**
>
> We thank the reviewer for their time reviewing our paper. We appreciate pointing out the novelties in our work and highlighting the importance of our contributions.
>
> > The main weakness of the paper is the lack of some key ablation studies, which make the claim that the method represents the scene better than the baselines less convincing.
> > - In addition to using local pixel entropy as input, the method also introduces several regularization terms in the training objective. It is possible that the improved performance comes from these new loss terms which are not included in the baseline models, they include:
>     1. Minimizing the norm of the distance traveled by each keypoint in the IT loss to regularizing excessive keypoints’ movement
>     2. The overlapping loss that encourages the keypoint to spread over the image
>     3. The additional activation status variable, and the corresponding active status loss that control (reduce if possible) the number of active keypoint
> > - These additional terms could be considered contributions of the paper, but they should be separated from the main claim that the information-based method is better than the CNN-based method. Therefore, two sets of experiments can be considered in the paper, (1) the ablation on the proposed method to show the effect of each regularization term, or (2) incorporating these terms in the baseline models when comparing them with the proposed method.
>
> We thank the reviewer for the valuable feedback. We agree that the ablation study in the previous version was minimal. In the new version of the paper, we conducted an extended ablation study of our model as per the suggestion (1) of the reviewer. We understand suggestion (2), but this would require extensive reworking of the baseline methods, hence, completely altering them, leading to completely different methodologies.
>
> In Table 1, we added MINT w/o Reg. as an ablation in the main paper, in which we only use the information losses, removing the movement regularization from the information transportation loss, the overlapping loss, and the active status loss. Additionally, in Appendix E.1. - page 23, we provide a more thorough ablation study, with both qualitative and quantitative results, and further discussion of these. Notably, we point the reviewer to Figures 10, 11, and 12 of the Appendix, where we show the role of the regularization terms, and to our ablation videos at https://sites.google.com/view/mint-iclr/ablations.
> We hope that these results cover the concerns raised by the reviewer.

---

> > ### Author Response · Authors · 2022-11-19
> > **Response to Reviewer Nich (2/2)**
> >
> > > - The qualitative results show a clear advantage of the proposed method which is very impressive. That is, for the proposed method, the movement of an object only affects the keypoints attached to its body, but for the baseline models, it also causes the movement of many keypoints surrounding the object. However, it is unclear if this behavior is due to architectural design, namely the receptive field and transporting resolution.
> >     1. Receptive filed. In the baseline model, e.g., transporter, the CNN features that the model learns to transport has a receptive field of 24 (input size is 64x64) for each position (computed based on the implementation the paper use [1]), so each pixel contains the information of a large area of the image. This might explain why the surrounding keypoints will also follow the movement of the object - they capture part of the object’s information. On the other hand, the image entropy is designed to represent local information, the entropy region size is three. Is it possible that the baselines can enjoy similar improvement with a careful receptive field design?
> >     2. Resolution. Based on transporter implementation [1], the resolution of the feature map to transport is 16x16, and the input image resolution is 64x64, both are smaller than the counterpart of the proposed method and could inevitably cause loss of information. Please provide some investigation into this factor.
> > [1] https://github.com/pairlab/v-cdn
> >
> > We thank the reviewer for this very interesting remark. Following this suggestion, we created Transporter-modified as a baseline. In the original implementation of the Transporter, the feature maps have a receptive field of size 24 for each position for an image input of size 128x128. The resolution of the feature maps between which the features are transported is 32x32. For fair comparison against our method, which uses an entropy region of size 3x3, we modified the network architecture to have a receptive field of 7 and a feature map of size 122x122. We call the new architecture Transport-modified. It is further explained in Appendix E.3 (page 26).
> >
> > The experimental results show that the Transporter-modified model outperforms the original Transporter in the quantitative evaluation of the CLEVRER dataset (Table 1 of the updated version). An interesting observation from the qualitative results in Figure 10 - page 26 is that we can see keypoints jumping from one object to another. We argue that this behavior is due to the smaller receptive field that leads the model to assign keypoints to features instead of objects, and thus keypoints jump to similar features in different objects. We hope that this experiment addresses the reviewer's concerns and clarifies the advantage of our method and contribution.
> >
> >
> > > - For reproducibility, the authors provide some of the hyper-parameters for the proposed model, but it is not sufficient to reproduce the experiment result. It is also not clear how to implement the parallelized entropy layer, it would be nice if the authors could release the code for future research.
> >
> > We want to refer the reviewer to our supplementary material. Both in the first submission and the updated one, the code is part of the supplemental material as a zip file. The code contains detailed instructions on how to set up the environments and reproduce the results, along with the cuda implementation of the entropy layer. We will release the code in a public GitHub, upon acceptance of the paper.
> >
> >
> > > - One small thing: It would be nice to provide a theoretical derivation on how to approximate a joint pixel-wise entropy as the max over the two marginals, i.e., how one can have $E(I_t,I_{t−1})=max(E(I_t),E(I_{t−1}))$
> >
> > We thank the reviewer for their suggestion. We added Lemma 1 in the main paper and provided the proof in Appendix C.2 - page 21.
> >
> >
> > We hope the reviewer finds our updates a significant improvement to our previous submission, particularly regarding the requested key experiments. If the reviewer has additional comments, please let us know.

---

> ### Author Response · Authors · 2022-12-07
> **Follow-up Response to Reviewer Nich**
>
> We thank the reviewer for their time checking our updated paper and increasing their score. Here are some additional clarifications.
>
> > The extra experiments on the additional loss terms for keypoint detection and tracking reveal that the information contributes significantly to the performance. In fact, in some cases, the model performs better without the additional loss terms, which is interesting. It would be nice to include this setting in other metrics and experiments as well.
>
> We ran experiment 2 (Learning dynamics) for MINT w/o Reg., and it shows superior performance as well:
>
> | Method | 1-step prediction | 2-step prediction | 3-step prediction |
> | :---:        | :---:   | :---:  | :---:        |
> | MINT w/o Reg. (ours) | 0.916 $\pm$ 0.063 | 0.907 $\pm$ 0.068 | 0.889 $\pm$ 0.076 |
>
> The results suggest that MINT w/o Reg. is a good option when economizing the number of keypoints is not important for the task and if the unsuccessful keypoint assignment does not confuse the system. Since our goal is to recover information with reasonable keypoint assignment, we found that the regularizers are important (see Figure 11 - Appendix E.1.).
>
> > Additionally, after reviewing the other reviewers' comments, I concur with Reviewer NNTd that the paper is theoretically overstated. For example, it is unclear to me how the proof in Appendix C.2 can justify the approximation .
>
> Appendix C.2. proves that the max of the two entropies is a lower bound for the joint entropy. The lower bound in equation (29) holds for any two random variables and is theoretically sound as per [1] section 6.1.3 - page 206. Following this general result, we present the pixel-wise lower-bound in equation (30), which we use to approximate the joint entropy image. We discussed why this approximation is acceptable in Appendix C.2.
>
> > Neither the assumption that two consecutive frames are independent
>
> The assumption of independence is mistakenly placed in the proof, the equations (29) and (30) hold for dependent random variables as well (as per [1]). We will remove this assumption from the proof, as correctly indicated by the reviewer. Moreover, if we can indeed assume independence, then the joint entropy equals the sum of the entropies, and we would not need to approximate it (this is also mentioned in [1] section 6.1.3 - page 206).
>
> [1] Murphy, K. P. (2022). Probabilistic machine learning: An introduction. MIT press.
> PDF is available on the webpage: https://probml.github.io/pml-book/book1.html
>
> > nor the statement that the joint will be equal to the max of two marginal are theoretically sound. It appears to be an empirical discovery rather than a result of mathematical analysis.
>
> Following the previous discussion, the lower bound of the joint entropy is a theoretically sound assumption. We decided to use the lower bound as an approximation of the joint entropy, and we designed our information-theoretic losses correspondingly. The empirical results support our decision to choose this approximation.
>
> We hope the provided responses resolve the concerns of the reviewer.

---

### Official Review · Reviewer_NNTd · 2022-10-26

**Confidence:** 3
**Correctness:** 2
**Technical Novelty And Significance:** 2
**Empirical Novelty And Significance:** 4
**Recommendation:** 3

**Clarity, Quality, Novelty And Reproducibility:**

Clarity:

The notations and the figures are unclear, it makes the paper hard to read
- "pixel-wise entropy image E(I) for an RGB input image I \in R^{HxWxC}". So the Entropy image is a function that takes an image and outputs an image R^{HxW}. But "local entropies E(I(x, y)) ..." suggests that the entropy image function takes individual pixels as inputs. eq1 is similarly confusing.
- The notations are overloaded in a confusing way, eg. eqn2, where the heatmap is indexed twice "h_i^(t)(x_i, y_i)". h could be considered as a heatmap function h(x_i^(t), y_i(t)), or as the heatmap image computed from the keypoint h_i^t, either would be good, but it is confusing now in the paper.
- On fig1 it is hard to see what is a function and what is a variable, especially between the "Masked entropy" and "M_a^(t)" image the arrows do not join, I am just guessing they are multiplied elementwise there. Also the losses could be show on fig1

Novelty:
For me using local entropy and training with the proposed loss is novel.

Quality:
The experimental results are strong, but there are severe problems withe the paper (see weaknesses)

Reproducibility:
Code is available, reproducing the results should not take too much effort


**Strength And Weaknesses:**

Strengths:

The method well on multiple tasks and performs better than the competing methods.

Weaknesses:

The title exaggerates the contributions: "An information-theoretic approach to unsupervised keypoint representation learning "
- The reader expects a theory that explains something not yet known, provide some optimality etc. There is no theorem, nor proof.
- When entropy is minimised or maximised, one wants to achieve some property for the associated random variable, and there is an explanation why that property is desired for the task. Here this is not the case the "local entropy" acts as a hand crafted saliency feature.

The explanation about the entropy is a forced analogy
- The losses and equations are not wrong, but the equations would make more sense if they described reconstructing a saliency pattern. E.g. CE(I_t|I_{t−1}) = max(E(I_t)-E(I_{t−1}), 0) could be interpreted as the novel saliency on the next frame and there is no need to approximate the joint entropy with the maximum.
- From a saliency perspective using heat maps make sense, the keypoints are used to reconstruct the pattern. Where do they come from an information theoretic perspective? All losses and explicite functional relationships (e.g. heatmaps) should be derived top-down from the theory and assumptions.

Some important content is delegated to the appendix, it could be better placed in the main text
- (E) Ablation study of the losses
- (C) Evaluation metrics. As this is listed as a contribution, this should be part of the main text

The ablation studies are limited. It would be important to see some results with different design choices, e.g. number of keypoints


**Summary Of The Paper:**

The paper presents a novel method for learning to represent images with keypoints, that can be used for many downstream tasks, like object detection and tracking and learning dynamics.
- A fixed number of keypoints are extracted from a neural network from each frame.
- An information transporter (similar to Kulkarni et al., 2019) is trained so the keypoints reconstruct local entropy patterns of neighboring frames
- novel performance metrics are introduced and the proposed method outpuerforms the competition on these metrics as well as on the standard metrics.

**Summary Of The Review:**

The paper is a mixed bag: strong results and bad presentation. There are two possibilities:
1) There is an theory, from which the losses can be derived. Then this is not presented, and the explanations are insufficient.
2) The local entropy is just a saliency feature. In this case the explanation is incorrect.

In both scenarios the paper is not ready for publication.

---

> ### Author Response · Authors · 2022-11-19
> **Response to Reviewer NNTd (1/2)**
>
> We thank the reviewer for their time and effort invested in reviewing our paper and for noting the novelty of our method in using local image entropy and the proposed losses.
>
> >When entropy is minimised or maximised, one wants to achieve some property for the associated random variable, and there is an explanation why that property is desired for the task. Here this is not the case the "local entropy" acts as a hand crafted saliency feature.
>
> We apologize if we presented our motivation and method in a convoluted way. We want to clarify possible misunderstandings with this answer and point the reviewer to the new additions in the paper, following their suggestion. In the following we discuss them in detail.
>
> In our work, we treat pixels as random variables, a common approach in computer vision, particularly in classical methods that use Markov random fields [1,2]. We use Shanon's definition of entropy to measure the information of pixels in an image. We make the original assumption that a keypoint is an entity representing information in a patch (its local heatmap). The reviewer is correct in that entropy can be used to measure saliency. Indeed, we also refer to local image entropy as a measure of intrinsic saliency (updated Sec.2 and in particular Sec 2.2.2 discuss this part). However, saliency is an event with different interpretations and can be measured in various ways. We added the extended related work section in Appendix G to explain better the use of spatial image entropy in computer vision and the use of information-theoretic losses in deep learning.
>
> [1] Stan Z Li. Markov random field modeling in image analysis. Springer Science & Business Media, 2009.
> [2] Q. R. Razlighi, N. Kehtarnavaz and A. Nosratinia. Computation of Image Spatial Entropy Using Quadrilateral Markov Random Field. IEEE TIP 2009.
>
> >The title exaggerates the contributions: "An information-theoretic approach to unsupervised keypoint representation learning "
>
> Regarding the title, we respectfully disagree since our method proposes novel losses based on well-established and known information-theoretic measures, as those are also explained, for example, in Ch. 6 of [3]. We use Shanon's entropy to measure local image information, which can be considered a measure of local saliency. Therefore, we rely on information-theoretic measures (i.e., entropy, conditional entropy, and mutual information) to derive a methodological framework for learning keypoint representations in videos that are spatiotemporally consistent.
> We hope that the new version of the paper and, in particular,  the updated Section 2, along with the extended related work discussion in App. G, clarify any misunderstandings regarding our motivation and the scope of our work.
>
> [3] Murphy, K.P., 2022. Probabilistic machine learning: an introduction. MIT press. url: https://probml.github.io/pml-book/book1.html
>
> >The reader expects a theory that explains something not yet known, provide some optimality etc. There is no theorem, nor proof.
>
> To further support the design of the proposed losses theoretically, we provide in the updated version in Section 2 bounds on the error probability of the information loss by the keypoint representation (proofs can be found in Appendix C - page 20).
> Further, we show with additional quantitative and qualitative results the efficacy of our proposed method, proving our original hypothesis that keypoints can be treated as transporters of information. We hope the reviewer finds the new version of the paper with the updated motivation discussion and theoretical findings a reasonable and significant improvement of our previous submission. We would be eager to clarify further concerns.

---

> > ### Author Response · Authors · 2022-11-19
> > **Response to Reviewer NNTd (2/2)**
> >
> > > The explanation about the entropy is a forced analogy
> > > - The losses and equations are not wrong, but the equations would make more sense if they described reconstructing a saliency pattern. E.g. CE(I_t|I_{t−1}) = max(E(I_t)-E(I_{t−1}), 0) could be interpreted as the novel saliency on the next frame and there is no need to approximate the joint entropy with the maximum.
> >
> > We use entropy to quantify local saliency and measure the amount of information transmitted by the keypoints using Shanon's entropy.
> > We mention in section 2.2, "We consider the keypoints as a compact representation of images, which attend to salient entities in a scene." Local entropy is a measure of intrinsic local saliency [4]. We use local pixel-wise entropy as a measure for local saliency, which in principle allows us to use information theory and its well-established measures for designing our proposed method and losses. To our best knowledge, there is no single way to compute saliency (e.g., some works use self-information [5], others use superpixel connectivity [6], others use conditional random fields [7]). Moreover, "conditional saliency" has been presented in [8] modeled through local conditional entropy. Our information-theoretic treatment allows us to approximate the conditional entropy using the max of the joint entropy, as noted also in Lemma 1  (Sec. 2 and proof in App. C.2) of  the updated paper. Accordingly, we use local conditional image entropy in a sequence of images to encourage keypoints to attend to moving objects.
> >
> > [4] Goferman, Stas,et al. "Context-aware saliency detection." IEEE TPAMI 2011.
> >
> > [5] Neil Bruce and John Tsotsos. Saliency based on information maximization. NeurIPS 2005.
> >
> > [6] Wangjiang Zhu, et al. Saliency optimization from robust background detection. CVPR 2014.
> >
> > [7] Le, T.N. and Sugimoto, A.,. Video salient object detection using spatiotemporal deep features. IEEE TIP 2018.
> >
> > [8] Li, Y., et al. Visual saliency based on conditional entropy. ICCV 2009.
> >
> > >From a saliency perspective using heat maps make sense, the keypoints are used to reconstruct the pattern. Where do they come from an information theoretic perspective? All losses and explicite functional relationships (e.g. heatmaps) should be derived top-down from the theory and assumptions.
> >
> > In our work, local image entropy represents the level of per-pixel information. Starting from the original assumption that a keypoint represents a patch of information on the image, we leverage local image entropy to measure the representation power of keypoints in terms of their transmitted amount of information.
> > The heatmap defines an area around a keypoint that allows the retrieval of local information. We learn our keypoint representation by extracting image features using as intrinsic training signal the spatiotemporal information coverage by the keypoints. From an information-theoretic perspective [9], the heatmap in our work can be assumed as the channel capacity around each keypoint. We updated Section 2 to clarify the motivation of our work. We hope the previous discussion and the updated submission clarifies any ambiguities in our previous paper presentation.
> >
> > [9] Shujian Yu, Luis G Sánchez Giraldo, and José C Príncipe. Information-theoretic methods in deep neural networks: Recent advances and emerging opportunities. IJCAI 2021.
> >
> > > Some important content is delegated to the appendix, it could be better placed in the main text
> >
> > We thank the reviewer for the constructive points. In the updated version, we note the ablation  MINT w/o Reg. in the main paper, and we additionally provide an extensive ablation study in the Appendix (Sec. E.1. page 23). We mainly ablate losses and regularizers used in our method. We show that if we know a priori the required number of keypoints (e.g., 10 for CLEVRER due to the simulated environment), the masked entropy loss alone is enough to perform well across all metrics (Table 6). However, we propose an approach where the number of keypoints should not be known a priori, and we rely on the auxiliary status loss to decide for the necessary number of keypoints to represent the scene.
> > Due to the page limit, we could not move the mathematical definition of the metrics and the whole ablation study into the main paper. We briefly describe the metrics in Sec. 3 - page 7. We strategically point the reader to the respective Appendix sections D and E for a detailed presentation of the metrics and the additional experimental results and ablations, respectively.
> >
> > > Clarity:
> >
> > We worked on improving the notation and clarity of the paper in the updated version. We updated Fig. 1 and 2 to show the operations more clearly, and we re-worked section 2, to be consistent with the notations.
> > We tried to address most of the reviewer's concerns in the updated version, and we hope we clarified possible misunderstandings, while shedding more light on the motivation and scope of our work. If the reviewer has additional comments, please let us know.

---

### Official Review · Reviewer_srpG · 2022-10-29

**Confidence:** 3
**Correctness:** 3
**Technical Novelty And Significance:** 3
**Empirical Novelty And Significance:** 2
**Recommendation:** 6

**Clarity, Quality, Novelty And Reproducibility:**

### Clarity
Mostly okay.
### Quality
Good
### Novelty
See comments in "Weaknesses".
### Reproducibility
I feel it should be good.

**Strength And Weaknesses:**

### Strength
* Unsupervised learning of keypoints is an important and difficult task. It avoids the requirement of extensive labels and can help many downstream tasks like matching, tracking etc.
* The proposed method learns reliable keypoint on both static and moving objects, and the number of active keypoints is compact.
* The proposed method works well on both synthetic objects and realistic complex scenarios.
### Weaknesses
* The novelty is not very strong. Some similar works on unsupervised saliency/keypoint learning have already been proposed and need to be discussed in this paper. For example, [a] has a segmentation network to predict masks (instead of keypoints) training in an unsupervised way, and [b] proposed to learn object parts in an unsupervised way. These two papers are not identical this the submission but share some similarities.
* The ME loss, the overlapping loss and the active status loss seem to work on single images instead of videos. I am curious how well it works on static images.


[a] SCOPS: Self-Supervised Co-Part Segmentation
[b] Interpretable Convolutional Neural Networks

**Summary Of The Paper:**

This paper proposes an unsupervised keypoint discovery method, especially for videos. To this end, authors propose to build an entropy layer to measure the information, and a deep learning model to predict keypoint. The model is learned in an unsupervised manner, by maximizing the information in a single frame and across frames. Authors demonstrate that the keypoints can be used for many downstream vision tasks such as detection, tracking and dynamic learning.

**Summary Of The Review:**

Overall I feel the paper above the acceptance bar, but there are still a bit more can be discussed and investigated, see in "Weaknesses".

---

> ### Author Response · Authors · 2022-11-19
> **Response to reviewer srpG**
>
> We thank the reviewer for taking the time to review our paper, pointing out the challenging aspects of unsupervised learning of keypoint representations, and highlighting the contributions and strengths of our empirical results.
>
> > The novelty is not very strong. Some similar works on unsupervised saliency/keypoint learning have already been proposed and need to be discussed in this paper.
> >
> We thank the reviewer for their constructive feedback. The novelty of our method lies in treating the keypoints as transporters of information. This treatment allows us to consider keypoints as a representation that ideally should cover all the information of the scenes in the video stream. This motivation led us to develop information-theoretic losses for unsupervised keypoint detection. We rephrased the introduction of Sec. 2 to motivate and highlight the originality of our approach better. To support our design of the proposed losses, we provide additional theoretical bounds w.r.t. the probability of error of “reconstructing” information using local entropy. We hope that the combination of the theoretical motivation and analysis, the strong empirical results, and the evaluation benchmark convince the reviewer and the community about the methodological novelty of our work.
>
>
> > For example, [a] has a segmentation network to predict masks (instead of keypoints) training in an unsupervised way, and [b] proposed to learn object parts in an unsupervised way. These two papers are not identical this the submission but share some similarities.
>
> We thank the reviewer for pointing us to this related work. We added both in our related work discussion. In particular, we would like to refer the reviewer to our Extended Related Work discussion in Appendix G of the updated paper pages 30-31, and the paragraphs "In that direction, (Hung et al., 2019; Zhang et al., 2018a) finetune pre-trained ImageNet features ... while we can economize the use of keypoints to adequately represent the video information with a minimum number of keypoints", where we also comment on the differences of our work w.r.t. [a,b].
>
> > The ME loss, the overlapping loss and the active status loss seem to work on single images instead of videos. I am curious how well it works on static images.
>
> We thank the reviewer for this interesting question. In Table 5 of Appendix E.1 in p. 25 of the updated paper, we added the suggested ablation, called MINT w/o Temp., which uses only losses operating on single images instead of video (i.e., the ME loss, overlapping loss, and status loss), and we provide additional qualitative results in the Appendix E.1. (Figures 10 and 12).
> The quantitative results in Table 5 show that MINT w/o Temp. can detect 85% of the objects in the scene while distributing the keypoint reasonably. Fig. 12 in p. 28 indicates that MINT w/o Temp. assigns keypoints to objects in the scene successfully. Fig. 10 shows that MINT w/o Temp. can detect the human and objects in the background, but due to the lack of temporal information, it does not concentrate on the moving objects (e.g.,  the hand of the human). Video results can also be found in https://sites.google.com/view/mint-iclr/ablations
>
>
> We hope the reviewer finds our updates a significant improvement to our previous submission, particularly concerning the reviewer's specific requests. If the reviewer has additional comments, please let us know.

---

### Author Response · Authors · 2022-11-19
**General response**

We thank all reviewers for their time and effort in providing constructive reviews. We hereby provide an updated version of our paper, where we addressed all raised points by the reviewers. We want to highlight that the method remained unchanged, as can be verified by the submitted code in the supplementary material. Specifically, in the updated version,
* we reformulated the overall motivation behind the information-theoretic treatment of our approach and rephrased several parts of the method to improve readability (Sec. 2);
* we clarified and fixed verbose notations and re-structured the explanation of the information transportation loss (Sec 2.2.2);
* we provide a theoretical analysis and lower bounds on the error probability of the image information represented by the keypoints to support better the design of the proposed losses (Sec. 2 and proofs in Appendix C - page 20);
* we added an extended related work discussion in Appendix G - page 30, to further highlight the novelty of work w.r.t. different methods in the literature;
* we provide an extensive ablation study with both qualitative and quantitative results (Sec 3 and extended analysis in Appendix E.1. - page 23);
* we created an additional baseline (Transporter-modified) by altering the receptive field of the original method (Table 1 and appendix E.3. - page 27).
* we noticed that the metric density of keypoints in an object (DAK) of the original version was misleading as it did not penalize the methods when no object was detected. We fixed this by altering this metric and renaming it to redundant keypoint assignment (RAK) (ref. Appendix D.4 - page 23). We updated the last metric in all experiments for all methods and ablations.

The project website https://sites.google.com/view/mint-iclr/home contains complementary videos from the different comparisons and ablations, and across the different tasks and datasets.

We hope the area chair and the reviewers find our updates a significant improvement to our previous submission.

---

### Author Response · Authors · 2022-12-02
**Feedback on rebuttal**

Dear reviewers and meta-reviewer,

according to your requests, we have updated the paper and tried to cover all your concerns. We would be grateful if you could acknowledge our rebuttal timely and allow us to answer possible remaining concerns. If you find our paper improved according to the suggestions, we would be glad if you considered raising your scores.

Thank you all for your service.

---

### Decision · Program_Chairs · 2023-01-20

**Decision:**

Reject

**Justification For Why Not Higher Score:**

My initial judgments from reading the discussion was that this was a quite likely straight accept, because it seemed to me from the discussion that the weaknesses pointed out by R2 were minor and picking somewhat minor nits, i.e., framing the method through the concept of saliency rather than information would motivate the method a bit more correctly.

But having taken a closer look at it, these are quite fundamental issues, and without them, the entire approach and its exposition are on rather shaky ground.

**Justification For Why Not Lower Score:**

N/A

**Metareview: Summary, Strengths And Weaknesses:**

This paper builds on the popular Transporter framework for keypoint representation learning, and trains keypoints such that, when transported to neighboring frames, they are able to accurately reconstruct "entropy images".

Strengths:
- The method is novel, and appears to work well on simulated images, and even on fairly complex real images which are known to be quite challenging for this line of work. And it is further evaluated for multiple tasks including imitation learning, often proposing new and reasonable metrics for more thorough evaluation, corresponding to more standard metrics.

Weaknesses:
- While results are strong, the conceptual motivation for the work is rather shaky, as pointed out by R2, and not satisfactorily addressed.
   - In the "entropy images" used throughout this work, a pixel is considered "high entropy" if it is the center of a small image patch with very diverse pixel values. This idea is not very well-motivated. Why should diverse pixel values in a small region be a sign of usefulness? Additionally, the "total information" of an image is very poorly approximated by the sum of pixelwise entropies, as defined above. This notion of information is very low-level, and does not correspond to any high-level semantic meaning such as objects, which is the putative goal of this work and all the methods it compares against. Rather, it harks back to older computer vision ideas such as Harris corner points. So why should "low-level information coverage" (corresponding to the first loss) and "low-level information transport" (corresponding to the second loss) yield anything more than corner points on moving objects?
   - The approximation of joint entropy as the higher of the two entropies is even shakier. First, it still treats image entropies as these low-level pixelwise entropies. Then, it further approximates joint entropy of (X1, X2) by the higher of the two entropies, of X1 and X2, justifying it through a loose bound. No conditions are stated for when / why this is a good approximation.

- On the experiments side, the methods it compares to are indeed representative of seminal keypoint detection work (Minderer et al 2019, and Transporter et al 2019), but they are both 3 years old, and it would be appropriate to attempt comparisons to more recent work.